# CRONOS: Enhancing Deep Learning with Scalable GPU Accelerated Convex Neural Networks

**Miria Feng**
Electrical Engineering
Stanford University
miria0@stanford.edu

**Zachary Frangella**
Management Science & Engineering
Stanford University
zfran@stanford.edu

**Mert Pilanci**
Electrical Engineering
Stanford University
pilanci@stanford.edu

## Abstract

We introduce the *CRONOS* algorithm for convex optimization of two-layer neural networks. CRONOS is the first algorithm capable of scaling to high-dimensional datasets such as ImageNet, which are ubiquitous in modern deep learning. This significantly improves upon prior work, which has been restricted to downsampled versions of MNIST and CIFAR-10. Taking CRONOS as a primitive, we then develop a new algorithm called CRONOS-AM, which combines CRONOS with alternating minimization, to obtain an algorithm capable of training multi-layer networks with arbitrary architectures. Our theoretical analysis proves that CRONOS converges to the global minimum of the convex reformulation under mild assumptions. In addition, we validate the efficacy of CRONOS and CRONOS-AM through extensive large-scale numerical experiments with GPU acceleration in JAX. Our results show that CRONOS-AM can obtain comparable or better validation accuracy than predominant *tuned* deep learning optimizers on vision and language tasks with benchmark datasets such as ImageNet and IMDb. To the best of our knowledge, CRONOS is the first algorithm which utilizes the convex reformulation to enhance performance on large-scale learning tasks.

## 1   Introduction

The non-convex landscape of deep neural networks (DNN) poses significant challenges for modern deep learning, especially since non-convexity implies it is NP-hard to train a neural network to optimality Blum and Rivest [1988]. Common stochastic first-order optimizers, such as stochastic gradient descent (SGD), offer no guarantees of producing more than an approximate stationary point Arjevani et al. [2023], which may be considerably suboptimal Ge et al. [2015]. The effectiveness of methods such as SGD and Adam are therefore reliant on heavily conditioned training environments. As a result, most models require extensive hyperparameter tuning of the optimizer to train successfully. This leads to expensive iterations in high compute settings with variable performance depending on optimizer selection and problem domain Yao et al. [2021]. Scaling laws Rosenfeld [2021] also indicate this regime will yield an increasingly larger number of hyperparameters, with a disproportionate dependence on computational resources and data cost.

Our objective is to achieve more efficient targeted optimization of deep learning tasks by leveraging the connection between neural networks and convex optimization. This approach provides clearer insight to the underlying optimization problem deep learning is trying to solve, despite the complexity of its non-convex landscape. The authors of Pilanci and Ergen [2020] have recently proven that the training of shallow neural networks can be equivalently formulated as a convex optimization program. The strategy leverages semi-infinite duality theory to develop algorithms which converge to the global minimum in polynomial time. However, the resulting convex program is a constrained high-dimensional linear model that is intensely difficult to solve at scale. Other more recent works of Mishkin et al. [2022] and Bai et al. [2023] have made progress in solving this problem with

38th Conference on Neural Information Processing Systems (NeurIPS 2024).

small downsampled versions of MNIST Noever and Noever [2021] and CIFAR-10 Krizhevsky and Hinton [2010], yet were unable to scale to realistic high-dimensional datasets that are ubiquitous in modern deep learning. This inability to handle large real-world data significantly limits the practical deployment of convex neural networks, despite their strong theoretical guarantees.

In this work, we propose CRONOS: the **C**onvex **R**eformulated **N**eural Network **O**perator **S**plitting algorithm. CRONOS is a fast, efficient and nearly hyperparameter-free method for training two-layer convex neural networks. We then augment our algorithm in CRONOS-AM: Cronos with Alternating Minimization, which extends applicability to neural networks of arbitrary architectures beyond just two-layer networks. We implement all experiments and algorithms in JAX Roth et al. [2024] with the RTX-4090 GPU. This allows CRONOS to fully leverage GPU acceleration and eliminate memory bottlenecks, thus successfully tackling problems with large data. In order to sustain strong convergence guarantees, parallelization, and robustness to hyperparameter tuning we utilize ADMM Boyd et al. [2011] as a core solver in our method. This enables exciting new time-efficient strategies for CRONOS to handle scalability in real world problems. We evaluate the performance of our algorithms in binary and multi-class classification tasks across a wide domain of large scale datasets (both image and language), on three model architectures (MLP, CNN, GPT-2). Additionally, our theoretical analysis proves the convergence of CRONOS to the global minimum of the convex reformulation under mild assumptions.

To the best of our knowledge, this is the first time convex reformulated neural networks have been successfully applied to large data such as ImageNet Recht et al. [2019] and large language modeling tasks with GPT-2 Budzianowski and Vulić [2019] architecture. Our main contributions can be summarized as follows [1]:

- We develop a practical algorithm via convex optimization and the Alternating Directions Method of Multipliers (ADMM) Boyd et al. [2011] to train two-layer ReLU neural networks with global convergence guarantees.
- Using this as a primitive, we extend our algorithm to effectively train multi-layer networks of arbitrary architecture by combining CRONOS with alternating minimization.
- We demonstrate the efficient practical applications of our method on real-world large scale image and language datasets, including ImageNet and IMDb.
- Our experiments are implemented in JAX - a functional programming paradigm for significant GPU acceleration. Results demonstrate performance speedups and successfully overcome the problem of memory bottlenecks to tackle large data.
- Our theoretical analysis proves the convergence guarantees to global minimum under mild assumptions of our proposed algorithms.

## 2 Related Work

CRONOS builds upon key ideas from previous literature on convex neural networks, ADMM, randomized numerical linear algebra and synergy with JAX implementation. Please see Appendix A for detailed discussions of related work according to each of these three areas of interest.

## 3 Background

This section introduces the convex reformulation of two-layer neural networks. We formalize this in two steps: **3.1)** foundational definition of convex ReLU neural networks and **3.2)** its equivalence as a linearly constrained Generalized Linear Model (GLM) with group lasso regularization.

### 3.1 Convex ReLU Neural Networks

Given a dataset $X \in \mathbb{R}^{n \times d}$, a two-layer ReLU multilayer perceptron (ReLU-MLP) with weights $W^{(1)} \in \mathbb{R}^{m \times d}, w^{(2)} \in \mathbb{R}^m$ outputs the prediction:

$$f_{W^{(1)}, w^{(2)}}(X) = \sum_{j=1}^{m} (XW_{1j})_+ w_{2j}. \tag{1}$$

---

[1]Our codebase is available at `https://github.com/pilancilab/CRONOS`

Here $(x)_+ = \max\{x, 0\}$ denotes the ReLU activation function.

Given targets $y \in \mathbb{R}^n$, the network in (1) is typically trained by minimizing the following non-convex loss function:

$$\min_{W_1, w_2} \ell\left(f_{W_1, w_2}(X), y\right) + \frac{\beta}{2} \sum_{j=1}^{m} ||W_{1j}||_2^2 + (w_{2j})^2, \tag{2}$$

where $\ell : \mathbb{R}^n \mapsto \mathbb{R}$ is the loss function, and $\beta \geq 0$ is the regularization strength. It is typically challenging to solve (2), since the optimizer often needs meticulous tuning of hyperparameters to ensure successful training. Such tuning is expensive, since it requires many iterations of running the optimizer across multiple hyperparameter configurations in a grid search to obtain good performance. This dramatically contrasts with the convex optimization framework, where algorithms come with strong convergence guarantees and involve minimal hyperparameters. Fortunately, it is possible to maintain the expressive capabilities of ReLU neural networks while still enjoying the computational advantages of convex optimization.

Pilanci and Ergen [2020] have shown (2) admits a convex reformulation, which elides the difficulties inherent in solving the deep learning non-convex landscape. This reformulation has the same optimal value as the original non-convex problem, provided $m \geq m^*$, for some $m \leq n + 1$. Therefore, reformulating (2) as a convex program does not result in loss of information or generality.

Pilanci and Ergen [2020]'s convex reformulation of (2) also enumerates the actions of all possible ReLU activation patterns on the data matrix $X$. These activation patterns act as separating hyperplanes, which essentially multiply the rows of $X$ by 0 or 1, and can be represented by diagonal matrices. For fixed $X$, the set of all possible ReLU activation patterns may be expressed as

$$\mathcal{D}_X = \left\{D = \text{diag}\left(\mathbb{1}(Xv \geq 0)\right) : v \in \mathbb{R}^d\right\}.$$

The cardinality of $\mathcal{D}_X$ grows as $|\mathcal{D}_X| = \mathcal{O}\left(r(n/r)^r\right)$, where $r := \text{rank}(X)$ Pilanci and Ergen [2020]. Given $D_i \in \mathcal{D}_X$, the set of vectors $v$ for which $(Xv)_+ = D_i Xv$, is given by the following convex cone:

$$\mathcal{K}_i = \{v \in \mathbb{R}^d : (2D_i - I)Xv \geq 0\}.$$

Learning directly based on the enumeration of $\mathcal{D}_X$ is impractical due to the exponential size of $\mathcal{D}_X$ [Mishkin et al., 2022]. Instead, we are motivated to work with the following convex program, based on sampling $P$ activation patterns from $\mathcal{D}_X$:

$$\min_{(v_i, w_i)_{i=1}^P} \ell\left(\sum_{i=1}^{P} D_i X(v_i - w_i), y\right) + \beta \sum_{i=1}^{P} ||v_i||_2 + ||w_i||_2 \tag{3}$$
$$\text{s.t. } v_i, w_i \in \mathcal{K}_i \quad \forall i \in [P].$$

Although (3) only works with a subsampled version of the convex reformulation of Pilanci and Ergen [2020], it can be shown under reasonable conditions that (3) still has the same optimal solution as (2) Mishkin et al. [2022]. Therefore we can confidently work with the tractable convex program in (3).

## 3.2 Convex ReLU networks as linearly constrained GLMs with group lasso regularization

Prior derivations Bai et al. [2023] have shown that (3) may be reformulated as a linearly constrained composite convex program:

**Proposition 3.1.** *Define the matrices $F_i = D_i X$ and $G_i = (2D_i - I)X$, where $i \in [P]$. Then by introducing the constraints $u_i = v_i, z_i = w_i$, where $i \in [P]$, and appropriate slack variables $s_1, \ldots, s_P, t_1, \ldots, t_P$, (3) can be reformulated as:*

$$\min_{(\boldsymbol{u}, \boldsymbol{v}, \boldsymbol{s})} \ell(F\boldsymbol{u}, y) + \beta||\boldsymbol{v}||_{2,1} + \mathbb{1}(\boldsymbol{s} \geq 0)$$
$$\text{s.t. } \begin{bmatrix} I_{2dP} \\ G \end{bmatrix} \boldsymbol{u} - \begin{bmatrix} \boldsymbol{v} \\ \boldsymbol{s} \end{bmatrix} = 0. \tag{4}$$

*where*

$$\boldsymbol{u}, \boldsymbol{v}, \boldsymbol{s} \in \mathbb{R}^{2dP}, F \in \mathbb{R}^{n \times 2dP}, G \in \mathbb{R}^{2nP \times 2dP}$$

The reformulation in (4) is essentially a very large *constrained* generalized linear model (GLM) with a group lasso penalty Yuan and Lin [2006]. The data matrix $F$ associated with (4) has dimensions

$n \times 2dP$, and the constraint matrix has dimensions $2(n + d)P \times 2dP$. Although the sizes of $F$ and the constraint matrix seem intractable, they are highly structured. Each $F_i$ and $G_i$ consists of the data matrix $X$ multiplied by a diagonal matrix of 0's and 1's. Therefore, $F$ and $G$ do not need to be instantiated, and we can apply matrix-vector products efficiently by exploiting this structure on GPU-accelerated frameworks. Additionally, if $X$ is approximately low-rank (a common phenomenon in machine learning), its singular values must decay fast. $F_i$ and $G_i$ then also inherit this approximate low-rank structure in (4), which CRONOS exploits and solves via fast matrix-vector products on GPU acceleration.

## 4 CRONOS: Convex Neural Networks via Operator Splitting

This section introduces the CRONOS algorithm for solving (4). CRONOS is a scalable, convex optimization-based learning method capable of handling large data and utilizes GPU acceleration for enhanced performance.

### 4.1 ADMM for robustness and decomposability

---
**Algorithm 1** ADMM for Convex ReLU Networks

---
**Require:** penalty parameter $\rho$
  **repeat**
    $u^{k+1} = \mathrm{argmin}_u \left\{ \frac{1}{2}\|Fu - y\|^2 + \frac{\rho}{2}\|u - v^k + \lambda^k\|_2^2 + \frac{\rho}{2}\|Gu - s^k + \nu^k\|^2 \right\}$
    $\begin{bmatrix} v^{k+1} \\ s^{k+1} \end{bmatrix} = \mathrm{argmin}_{v,s} \beta\|v\|_{2,1} + \mathbb{1}(s \geq 0) + \frac{\rho}{2}\|u^{k+1} - v + \lambda^k\|^2$     ▷ Primal update
    $\lambda^{k+1} \leftarrow \lambda^k + \frac{\gamma_\alpha}{\rho}(u^{k+1} - v^{k+1})$     ▷ Dual $\lambda$ update
    $\nu^{k+1} \leftarrow \nu^k + \frac{\gamma_\alpha}{\rho}(Gu^{k+1} - s^{k+1})$     ▷ Dual $\nu$ update
  **until** convergence

---

Equation (4) is in a natural form to apply the Alternating Directions Method of Multipliers (ADMM), the seminal first-order optimization algorithm for solving convex optimization problems Boyd et al. [2011]. ADMM is an ideal choice due to its strong convergence guarantees, robustness to hyperparameter variations, and ability to effectively leverage modern computing architectures through parallelism. However, directly applying ADMM to (4) requires solving the $u$-subproblem at each iteration:

$$u^{k+1} = \mathrm{argmin}_u \{ \frac{1}{2}\|Fu - y\|^2 + \frac{\rho}{2}\|u - v^k + \lambda^k\|_2^2 + \frac{\rho}{2}\|Gu - s^k + \nu^k\|^2 \}. \qquad (5)$$

This solution then requires solving the following linear system:

$$
\begin{aligned}
(H + I)u^{k+1} &= b^k, \\
H &= \frac{1}{\rho}F^T F + G^T G, \\
b^k &= \frac{1}{\rho}F^T y + v^k - \lambda^k + G^T \left( s^k - \nu^k \right).
\end{aligned}
\qquad (6)
$$

Since $F$ is a $n \times 2dP$ matrix and $G$ is a $2nP \times 2dP$ matrix, the cost of solving (6) via a direct method is $\mathcal{O}(nd^2P^4 + d^3P^3)$. This is prohibitively expensive since $n$ and $dP$ are typically large. A natural method is to solve (6) inexactly via the Conjugate Gradient (CG) algorithm, which only requires matrix-vector products (matvecs) with $F, F^T, G$ and $G^T$. Under these conditions ADMM will still converge, since the sequences of subproblem errors are summable Eckstein and Bertsekas [1992]. However the number of iterations required by CG to achieve $\delta$-accuracy is $\mathcal{O}(\kappa(H + I)\log(1/\delta))$, where $\kappa(H + I)$ is the condition number of $H + I$. This results in slow convergence for machine learning problems, since $F$ and $G$ are closely related to the data matrix $X$, which is approximately low-rank Udell and Townsend [2019]. Given that the linear system in (6) must be solved at each iteration, CG's slow convergence renders it infeasible for real-world application.

## 4.2 Nyström preconditioning for fast convergence

---

**Algorithm 2** CRONOS

---

**Require:** penalty parameter $\rho$, forcing sequence $\{\delta^k\}_{k=1}^{\infty}$, rank parameter $r$

$[U, \hat{\Lambda}] = \text{RandNyströmApprox}(F^T F + G^T G, r)$        ▷ Compute using Algorithm 3

  **repeat**

    Use Nyström PCG (Algorithm 4) to find $\boldsymbol{u}^{k+1}$ that solves (5) within tolerance $\delta^k$

    $\boldsymbol{v}^{k+1} \leftarrow \textbf{prox}_{\frac{\beta}{\rho}\|\cdot\|_2}(\boldsymbol{u}^{k+1} + \lambda^k)$        ▷ Primal $\boldsymbol{v}$ update

    $\boldsymbol{s}^{k+1} \leftarrow (G\boldsymbol{u}^{k+1} + \nu)_+$        ▷ Slack $\boldsymbol{s}$ update

    $\lambda^{k+1} \leftarrow \lambda^k + \frac{\gamma_\alpha}{\rho}(\boldsymbol{u}^{k+1} - \boldsymbol{v}^{k+1})$        ▷ Dual $\lambda$ update

    $\nu^{k+1} \leftarrow \nu^k + \frac{\gamma_\alpha}{\rho}(G\boldsymbol{u}^{k+1} - \boldsymbol{s}^{k+1})$        ▷ Dual $\nu$ update

  **until** convergence

---

We exploit the approximate low-rank structure in our problem matrices to speed up convergence by applying the NysADMM algorithm from Zhao et al. [2022], which is an ADMM-based method targeted at solving large machine learning tasks. Notably, the subproblem for $\boldsymbol{v}^{k+1}$ and $\boldsymbol{s}^{k+1}$ has a closed-form solution that may be computed in $\mathcal{O}\left((n+d)P\right)$ time. NysADMM employs the Nyström Preconditioned Conjugate Gradient (NysPCG) from Frangella et al. [2023a] to solve (6). NysPCG is a linear system solver specializing in solving linear systems with large, approximately low-rank matrices.

NysPCG first constructs a low-rank preconditioner $P$ for the matrix $H + I$ in (6) from a *randomized Nyström approximation* of $H$. When $P$ is constructed with the appropriate rank, it can be shown that Nyström PCG solves the linear system in (6) to $\delta$-accuracy within $\mathcal{O}(\log(1/\delta))$ iterations (Proposition 6.3). The dependence on the condition number is therefore eliminated, and (5) can be solved quickly at each iteration. Details of the NysPCG algorithm and construction of the preconditioner are presented in Appendix B.1.

We refer to our algorithm in solving (4) as *CRONOS* (**C**onvex **R**eLU **O**ptimized **N**eural networks via **O**perator **S**plitting). The CRONOS algorithm is presented in Algorithm 2.

### 4.3 Scale and speed with JAX and Just-In-Time compilation

Scaling convex neural networks to realistic high-dimensional data is critical for machine learning problems. Therefore we implement our methods in JAX Bradbury et al. [2018], a high-performance numerical computing library designed to accelerate machine learning research. This framework provides an efficient way to perform array operations, automatic differentiation, and optimization of numerical processes. Leveraging Just-In-Time (JIT) compilation capabilities through XLA (Accelerated Linear Algebra) allow us to execute optimized machine code with extremely accelerated and scalable performance on GPU. Currently, the importance of GPUs in deep learning cannot be overstated. Therefore we note that any practically competitive algorithms need to fully utilize parallel processing capabilities. The combination JAX and JIT compilation effectively enables us to scale to high-dimensional datasets such as Food ($267 \times 267 \times 3$), ImageNet ($512 \times 512 \times 3$), and the IMDb language dataset. Our experiments are summarized in Section 7.

## 5 CRONOS-AM: Deep Networks via Alternating Minimization

In this section, we introduce the CRONO-AM algorithm for training a neural network with arbitrarily many layers. Consider training an $L$-layer neural network with ReLU activations and the least-squares loss. Training the network involves solving:

$$\underset{\theta}{\text{minimize}} \quad \|\mathcal{F}(\theta; X) - y\|^2 + \frac{\alpha}{2}\sum_{i=1}^{L-2}\|W_i\|_F^2 + \frac{\beta}{2}\left(\|W_{L-1}\|_F^2 + \|w_L\|^2\right),$$

where $\theta = (\text{vec}(W_1), \text{vec}(W_2), \dots, w_L) \in \mathbb{R}^p$ is the vector of the network weights, with $W_i$ denoting the weights for the $i$th layer, while $X$ is the data matrix, and $y$ are the labels. We can rewrite

this objective as:

$$\underset{\theta_{1:L-2},W_{L-1},w_L}{\text{minimize}} \left\| \left( \mathcal{F}_{1:L-2}(\theta_{1:L-2}, X)W_{L-1}^T \right)_+ w_L - y \right\|^2 + \frac{\alpha}{2} \sum_{i=1}^{L-2} \|W_i\|_F^2 + \frac{\beta}{2} \left( \|W_{L-1}\|_F^2 + \|w_L\|^2 \right),$$

where $\theta_{1:L-2} = (\text{vec}(W_1), \text{vec}(W_2), \dots \text{vec}(W_{L-2}))$, $\mathcal{F}_{1:L-1}$ consists of the first $L - 2$ layers. Let us write $\tilde{X}(\theta_{1:L-2}) = \mathcal{F}_1(\theta_{1:L-2}, X)$, which may be viewed as a transformed data matrix. We shall sometimes write $\tilde{X}$ for brevity. Hence, we can write the output of the network as:

$$\mathcal{F}(w; X) = \left( \tilde{X}(\theta_{1:L-2})W_{L-1}^T \right)_+ w_L.$$

Thus, the training problem may be rewritten as:

$$\underset{\theta_{1:L-2},w_{L-1},w_L}{\text{minimize}} \left\| \left( \tilde{X}(w_{1:L-2})W_{L-1}^T \right)_+ w_L - y \right\|^2 + \frac{\alpha}{2} \sum_{i=1}^{L-2} \|W_i\|_F^2 + \frac{\beta}{2} \left( \|w_{L-1}\|^2 + \|w_L\|^2 \right).$$

When $\theta_{1:L-2}$ is fixed, the preceding problem can be viewed as training a two-layer ReLU neural network with transformed data matrix $\tilde{X}$. Motivated by this, we replace the part of the optimization involving the last two layers with the convex reformulation to obtain:

$$
\begin{aligned}
&\underset{(\boldsymbol{u},\boldsymbol{v},\boldsymbol{s}),\, w_{1:L-1}}{\text{minimize}} \quad \|F\left( \tilde{X}(w_{1:L-1}) \right) \boldsymbol{u} - y\|^2 + \tfrac{\alpha}{2} \sum_{i=1}^{L-2} \|W_i\|_F^2 + \beta \|\boldsymbol{v}\|_{2,1} + \mathbb{1}(\boldsymbol{s} \geq 0). \\
&\text{s.t.} \begin{bmatrix} I_{2dP} \\ G \end{bmatrix} \boldsymbol{u} - \begin{bmatrix} \boldsymbol{v} \\ \boldsymbol{s} \end{bmatrix} = 0.
\end{aligned}
\tag{7}
$$

Equation (7) decouples the convex weights from the non-convex weights. This puts the objective into a natural form to apply alternating minimization, where we alternate between minimizing with respect to $w_{1:L-2}$ and $(\boldsymbol{u}, \boldsymbol{v}, \boldsymbol{s})$. To handle the convex minimization, we can apply CRONOS. For the non-convex portion, we propose utilizing DAdapted-Adam [Defazio and Mishchenko, 2023], as it does not require setting the learning rate. we call this algorithm CRONOS-AM (CRONOS-Alternating Minimization). Pseudocode for CRONOS-AM is given in Algorithm 5.

## 6 Theoretical Analysis of CRONOS

In this section, we establish the rate of convergence for CRONOS in solving (4), and provide an overall computational complexity estimate. Additionally, we show that the linear system for the $\boldsymbol{u}^k$-update may be solved by PCG at a rate independent of the condition number.

### 6.1 $F_i's$ and $G_i's$ inherit approximate low-rank structure of $X$

We begin by showing the spectrum of $X^T X$ upper bounds the spectrum of $F_i^T F_i$ and $G_i^T G_i$

**Proposition 6.1** ($F_i$ and $G_i$ are approximately low-rank if $X$ is). *Let $i \in [P]$, and $j \in [d]$. Then the eigenvalues of $F_i^T F_i$ and $G_i^T G_i$ satisfy:*

$$\max\{\lambda_j(F_i^T F_i), \lambda_j(G_i^T G_i)\} \leq \lambda_j(X^T X).$$

Proposition 6.1 shows that if $X$ is an approximately low-rank matrix, so are all the $F_i$'s and $G_i$'s. Most data matrices in machine learning exhibit fast polynomial or exponential spectral decay and so are well-approximated by low-rank matrices Wainwright [2019], Derezinski et al. [2020]. As the matrix $H$ that defines the linear system arising from (5) is built from the $F_i's$ and the $G_i's$, it will also be approximately low-rank, which motivates the following assumption.

**Assumption 6.2** (Approximate low-rank structure). Let $\lambda_j(H)$ denote the $jth$ eigenvalue of $H$. Then the eigenvalues values of $H$ satisfy:

$$\lambda_j(H) = \mathcal{O}\left( j^{-2\beta} \right), \quad \beta > 1/2.$$

Assumption 6.2 posits that the eigenvalues of $H$ decay at a polynomial rate, which is reasonable given the preceding discussion. Under Assumption 6.2, we will obtain complexity guarantees that align with CRONOS' practical performance.

## 6.2 Fast $u^k$-update

The next proposition shows that NysPCG solves the linear system in each iteration in (5) fast.

**Proposition 6.3** (Fast solution of $u$-subproblem). *Assume Assumption 6.2 holds. Suppose the randomized Nyström preconditioner is constructed with rank $r = \mathcal{O}\left(\log(\frac{1}{\zeta})\right)$. Then after $t = \mathcal{O}\left(\log(\frac{1}{\delta})\right)$ iterations, Nyström PCG outputs a point $\boldsymbol{u}^{k+1}$ satisfying*

$$\|(H + I)\boldsymbol{u}^{k+1} - b^k\| \leq \delta,$$

*with probability at least $1 - \zeta$.*

For fixed $\zeta > 0$, Proposition 6.3 shows that with a rank of $r = \mathcal{O}(1)$, NysPCG solves the $\boldsymbol{u}$-subproblem within $\mathcal{O}\left(\log(\frac{1}{\delta^k})\right)$ iterations. Thus, PCG solves the linear system quickly. Proposition 6.3 is consistent with practice, NysPCG with rank of $r = 20$ and 10-40 PCG iterations work well across all problem instances. Therefore solving (5) is not a bottleneck for CRONOS.

## 6.3 Convergence of CRONOS

The following theorem shows CRONOS converges to the global minimum of (4).

**Theorem 6.4** (Convergence and Computational Complexity of CRONOS). *Suppose Assumption 6.2 holds. Fix $\zeta \in (0, 1)$ and denote the minimum of (4) by $p^\star$. Construct the Nyström preconditioner in Algorithm 2 with rank $r = \mathcal{O}\left(\log(\frac{1}{\zeta})\right)$, and at the $k$th CRONOS iteration run Nyström PCG with tolerance $\delta^k$ to convergence. Then, with probability at least $1 - \zeta$ the following statements hold:*

*1. After $K$ iterations, CRONOS' output satisfies*

$$\ell(F\bar{\boldsymbol{u}}^K, y) + \beta\|\bar{\boldsymbol{v}}^K\|_{2,1} + \mathbb{1}(\bar{\boldsymbol{s}}^K \geq 0) - p^\star = \mathcal{O}(1/K),$$

$$\left\|\begin{bmatrix} I_{2dP} \\ G \end{bmatrix} \bar{\boldsymbol{u}}^K - \begin{bmatrix} \bar{\boldsymbol{v}}^K \\ \bar{\boldsymbol{s}}^K \end{bmatrix}\right\| = \mathcal{O}(1/K).$$

*2. The total complexity of CRONOS to produce an $\epsilon$-suboptimal point of (4) is*

$$\mathcal{C}_{\text{CRONOS}} = \tilde{\mathcal{O}}\left(\frac{ndP^2}{\epsilon}\right)$$

Theorem 6.4 shows CRONOS converges ergodically at an $\mathcal{O}(1/K)$-rate. When the minimum of (4) and (2) coincide[2], Theorem 6.4 guarantees CRONOS is within $\epsilon$ of the *global minimum* of (2) after $\mathcal{O}(1/\epsilon)$ iterations. By comparison, in the worst-case SGD on (2) can only be guaranteed to output an $\epsilon$-approximate stationary point after $\mathcal{O}(1/\epsilon^4)$ iterations Arjevani et al. [2023]. SGD's output may also fail to be close to a local minimum, and can be highly suboptimal Ge et al. [2015]. Therefore, CRONOS offers much stronger convergence guarantees than SGD.

Theorem 6.4 is the first realistic convergence guarantee for ADMM on (4). Previously the convergence analysis of Bai et al. [2023] assumes the ADMM subproblems are solved exactly at each iteration, which is unlikely for large-scale data. Consider a precision $\epsilon > 0$, then the cost of the state-of-the-art ADMM approach from Bai et al. [2023], is at least $\mathcal{O}(nd^2P^4 + d^3P^3)$ since it solves the linear system derived from Eq. (5) exactly. In contrast, the dependence of CRONOS upon $n, d$, and $P$ offers a significant improvement, since the cost grows linearly with $n$ and $d$. This enables our scalability to very high-dimensional datasets in both vision and language.

Theorem 6.4 as presented is pessimistic about the overall convergence speed of CRONOS. For sparsity-promoting convex regularizers such as the group lasso penalty, ADMM is known to reach the manifold containing the support in finite time, after which it converges linearly Liang et al. [2017], Yuan et al. [2020]. Therefore, CRONOS convergence is practically much faster than what Theorem 6.4 suggests.

---

[2]For a detailed discussion when this occurs, please see Appendix D.

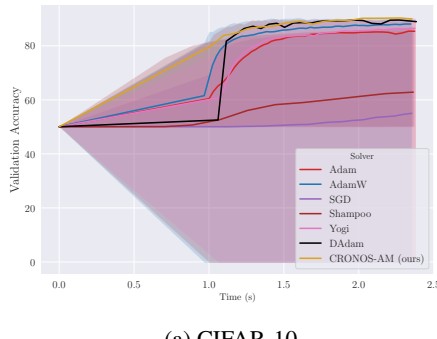
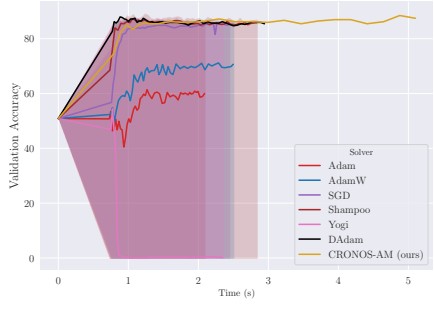

| (a) CIFAR-10 | (b) ImageNet |
|---|---|

Figure 1: CRONOS-AM vs. competitors on Deep ReLU MLP

# 7 Experiments

In this section, we empirically evaluate the efficacy of CRONOS and CRONOS-AM on classification tasks. We first experiment with vision data in multi-layer perceptron (MLP) and convolutional neural network (CNN) architectures[3]. We find that without the necessity of tuning hyperparameters, CRONOS performs as well or better than prevalent standard optimizers. To the best of our knowledge, CRONOS is the first convex reformulated neural network capable of handling large scale data tasks such as ImageNet classification. Secondly, we evaluate the performance of CRONOS on natural language sentiment classification with GPT2 architecture. All experiments were run in JAX v0.4.28 and FLAX v0.8.2. Appendix F presents detailed discussion of the experimental setup and additional numerical results. Details on each dataset used can be found in Appendix F.3.

## 7.1 Training a deep Multi-Layer Perceptron

We evaluate the performance of CRONOS-AM for binary classification with a 4-layer multilayer perception whose architecture is provided in the Appendix F.5. We train the model on three different datasets CIFAR-10, Food, and ImageNet [4]. CRONOS-AM is benchmarked against several of the most popular optimizers in deep learning: SGD with Polyak momentum (SGD) [Sebbouh et al., 2021], Adam [Kingma and Ba, 2014], AdamW [Loshchilov and Hutter, 2017], Shampoo [Gupta et al., 2018], Yogi [Zaheer et al., 2018] and D-adapted Adam (DAdam) [Defazio and Mishchenko, 2023]. For each competing method we consider 5 learning rates, selected randomly from a logarithmic grid with range $[10^{-5.5}, 10^{-1.5}]$.

Fig. 1 plots the median trajectory across different learning rates of each competing method along with the 5th and 95th quantiles. Fig. 1 shows CRONOS-AM either achieves the best or comparable performance relative to its competitors. These plots and the tables show competing methods exhibit an extremely high degree of variance, and poor learning rate selections can yield non-convergent behavior. In contrast, CRONOS-AM does not exhibit these weaknesses and performs comparably to the best-tuned competitor.

Table 1: Results for CIFAR-10 and ImageNet Datasets

| Optimizer | CIFAR-10 | | ImageNet | |
|---|---|---|---|---|
| | Peak Validation Range | Best Learning Rate | Peak Validation Range | Best Learning Rate |
| CRONOS-AM | 90.5% | NA | 88.47% | NA |
| DAdam | 90.15% | NA | 87.96% | NA |
| Adam | $[50.4, 89.8]\%$ | $3.79 \times 10^{-5}$ | $[50.87, 88.47]\%$ | $3.68 \times 10^{-6}$ |
| AdamW | $[50.04, 90.25]\%$ | $1.56 \times 10^{-4}$ | $[50.87, 87.46]\%$ | $4.07 \times 10^{-6}$ |
| Yogi | $[50.04, 90.84]\%$ | $5.10 \times 10^{-3}$ | $[50.87, 88.47]\%$ | $6.65 \times 10^{-6}$ |
| SGD | $[50.04, 87.75]\%$ | $5.71 \times 10^{-3}$ | $[50.87, 87.46]\%$ | $4.94 \times 10^{-5}$ |
| Shampoo | $[51.5, 89.15]\%$ | $5.07 \times 10^{-3}$ | $[51.5, 89.72]\%$ | $1.70 \times 10^{-4}$ |

---

[3]The results for the convolutional experiments may be found in Appendix F.6

[4]Results for Food and other downsampled variants of ImageNet may be found in Appendix F.5

Table 2: Optimizer Runtimes (s) on CIFAR-10 and ImageNet

| Dataset | CRONOS-AM | Adam | AdamW | D-Adapted Adam | SGD | Shampoo | Yogi |
|---------|-----------|------|-------|----------------|-----|---------|------|
| CIFAR-10 | 3.00 | 3.02 | 3.14 | 3.68 | 2.28 | 6.80 | 2.79 |
| ImageNet | 5.10 | 1.84 | 3.14 | 2.94 | 2.48 | 5.19 | 1.98 |

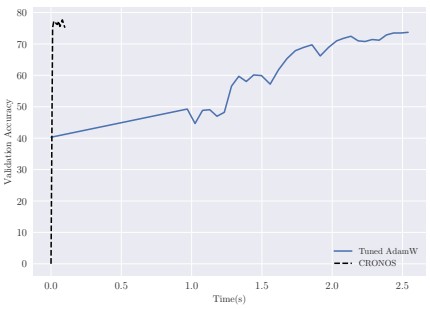

(a) CRONOS vs. tuned AdamW on GPT2-NFT

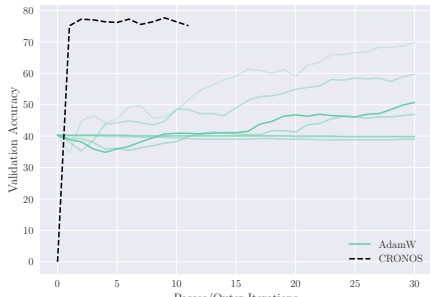

(b) CRONOS vs. AdamW trajectories on GPT2-NFT.

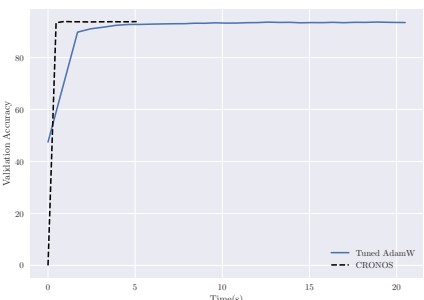

(c) CRONOS vs. tuned AdamW on GPT2-FT.

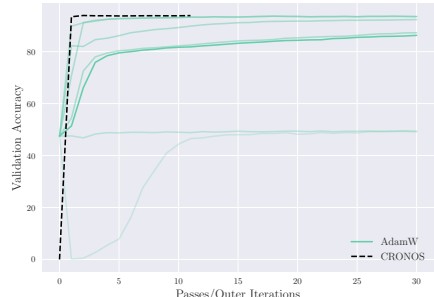

(d) CRONOS vs. AdamW trajectories on GPT2-FT.

Figure 2: CRONOS vs. AdamW on two GPT2 configurations for IMDb

Table 1 presents the range in peak validation across the grid (except for CRONOS-AM and DAdam, which do not require a learning rate parameter). CRONOS-AM outperforms DAdam on both tasks and performs comparably to the best-tuned first-order optimizer. On ImageNet, Shampoo does best by a fair margin. We attribute this to Shampoo being an approximate second-order optimizer. Properly tuned, Shampoo may yield better performance than purely first-order optimizers like CRONOS-AM and Adam for certain tasks.

Table 2 shows the total runtime in seconds for each optimizer on CIFAR-10 and ImageNet. Despite doing more work than competing optimizers, CRONOS-AM's runtime is comparable with standard optimizers such as Adam, AdamW, and Shampoo.

## 7.2 Natural Language Classification with CRONOS

Table 3: Results for Different GPT2 Architectures

| | GPT2-NFT | | GPT2-FT | |
|---|---|---|---|---|
| Optimizer | Peak Validation Range | Best Learning Rate | Peak Validation Range | Best Learning Rate |
| CRONOS | 77.66% | NA | **93.91%** | NA |
| AdamW | $[40.29, 73.69]\%$ | $1.56 \times 10^{-3}$ | $[48.30, 93.69]\%$ | $1.32 \times 10^{-4}$ |

Our experiments explore sentiment classification with the IMDb dataset. For all experiments, we use the pretrained GPT2 architecture with 12 transformer blocks and an embedding dimension of 768. We use the same GPT2Tokenizer across all language model tasks to ensure consistent evaluation,

and all dataset examples are padded to the same length of 60 words. The input to CRONOS is thus shaped into 9 batches of size (1500 x 60 x 768) for each of the 2 labels, and all experiments implement multiple trials across varying data batches. Our baseline benchmark model is the GPT2 pretrained model passed through one dense linear layer for classification accuracy. Numerical results are summarized in Table 3, and Fig. 2 compares CRONOS to tuned AdamW on time. CRONOS is seen to reach best validation faster than AdamW, particularly in Fig. 2a. Fig. 2b and Fig. 2d plot several AdamW trajectories along with CRONOS against a number of epochs for AdamW and ADMM iterations for CRONOS. The more translucent the curve for AdamW indicates larger deviation from median trajectory. Both plots show AdamW is extremely sensitive to the learning rate selection. Appendix F.4 provides further details of the three CRONOS integrated GPT2 experiments.

**IMDb-NFT.** We extract the GPT2 pretrained checkpoint and then immediately follow up with CRONOS for sentiment classification. Results are shown in Fig. 2a and Fig. 2b. Notably, this setting does not involve any training loops with standard optimizers (such as AdamW), and the foundation GPT2 model does not see any labeled sentiment data from the IMDb dataset. We limit the amount of data seen by CRONOS to only two batches to evaluate the efficacy of our method in the low data regime. Table 3 shows CRONOS significantly outperforms tuned AdamW. It reaches higher validation accuracy faster than AdamW and has the benefit of not requiring hyperparameter grid search. In contrast, Fig. 2 shows AdamW's performance may be quite poor if the learning rate is not judiciously selected.

**IMDb-FT.** Our fine-tuned experiment setting utilizes the GPT2 pretrained checkpoint followed by *one epoch only* of training with the AdamW optimizer on default parameters with the standard dense linear layer head followed by CRONOS. Results are shown in Fig. 2c and Fig. 2d. Although the authors of BERT Devlin [2018] recommend 2-4 epochs for training in this setting, we aim to evaluate the performance of CRONOS with limited overhead to extract features for maximum efficiency. The summary of results in Table 3 are particularly promising — CRONOS reaches **93.91%** validation accuracy, about 0.3% better than tuned AdamW, which is widely regarded as the most effective method for training language models.

**IMDb-DA.** We examine the setting of training for one epoch of AdamW on *unlabeled* IMDb data initialized from the GPT2 checkpoint. The resulting features are then extracted and passed into CRONOS for classification. The motivation for the experiment is to examine the potential of CRONOS on unsupervised domain adaptation settings in future work, which aims to reduce the distribution gap between source and unlabeled target domains. Our goal is to leverage the pre-trained high-level semantic knowledge in GPT2 and use the unlabeled IMDB data to help align this into our sentiment classification setting. Comprehensive experimental results are further summarized in Appendix F, and show promising directions for future work.

# 8   Conclusion

We introduce CRONOS, the first algorithm to successfully apply convex neural networks to high-dimensional datasets with competitive performance. CRONOS leverages ADMM for robustness and parallelization, Nyström preconditioning for fast convergence, and JAX for GPU acceleration with large data. We extend this framework with CRONOS-AM: a novel algorithm which utilizes alternating minimization to adapt convex reformulation for deeper networks of arbitrary architecture. CRONOS comes with strong theoretical guarantees that match its practical performance, a rarity for optimization algorithms in deep learning. Experiments on large benchmark datasets in image and language tasks validate the efficacy of our algorithms. CRONOS performs better or comparably against other prevalent optimizers, but with virtually zero tuning of hyperparameters. We achieve validation accuracy of **88.47%** on ImageNet binary classification, and **93.91%** on IMBd sentiment classification. Our goal is to advance an innovative, alternative paradigm in deep learning through convex networks, thus enhancing both efficiency and interpretability.

Our results raise several important questions for future work and present many exciting avenues for continued research. Can we provide a convergence guarantee for CRONOS-AM that shows an advantage over stochastic first-order methods? Additionally, the efficacy of CRONOS on NLP tasks suggest that investigating the performance of CRONOS on harder language datasets is an interesting direction, with potential for scalability on multi-GPU or TPU settings with other modalities.

**Acknowledgments.** This work was supported in part by National Science Foundation (NSF) under Grant DMS-2134248; in part by the NSF CAREER Award under Grant CCF-2236829; in part by the U.S. Army Research Office Early Career Award under Grant W911NF-21-1-0242; in part by the Office of Naval Research under Grant N00014-24-1-2164. We thank Lucy Woof and Peter Hawkins for support throughout and many insightful discussions.

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

# A   Related Work

**Convex Neural Networks.** Given the recent impressive capabilities of neural networks, there has been much work to develop strategies to avoid certain computational difficulties inherent in non-convex optimization. Bengio et al. [2005] is an early work in this direction; they treat 2-layer neural networks as convex models but require the first layer to be fixed. Random features regression is a similar approach, where the first layer is randomly initialized, and only the last layer is trained Rahimi and Recht [2008]. More recently, it has been shown for certain classes of wide neural networks that training is equivalent to solving a convex kernel regression problem Jacot et al. [2018]. Wide neural networks of this form are said to be in the "kernel regime". However although random feature regression and kernel regime neural networks yield convex optimization setups, they are known to exhibit inferior performance to standard neural networks Chizat et al. [2019], Ghorbani et al. [2021]. This makes them less attractive relative to the convex reformulation of (3).

Motivated by the strong optimality guarantees associated with the convex reformulation (3), other lines of work directly try and solve (3) using techniques from convex optimization. Two relevant works and the current state-of-the-art, are Mishkin et al. [2022] and Bai et al. [2023]. Mishkin et al. [2022] develops a variant of restarted-FISTA (r-FISTA) Nesterov [2013], Odonoghue and Candes [2015] to solve (3), with experiments on MNIST and CIFAR10. Although CRONOS and Mishkin et al. [2022]'s r-FISTA have similar theoretical complexity profiles, we observe that CRONOS is able to outperform r-FISTA in terms of scalability, as it is able to handle large datasets ImageNet and IMDb. In contrast, the $r$-FISTA solver of Mishkin et al. [2022] was unable to scale to all of CIFAR-10 and MNIST. More closely related is the work of Bai et al. [2023], who directly applies ADMM to (4). However this strategy is unsuitable for large problems as it requires forming and factoring a sizable matrix. CRONOS alleviates this issue by applying NysADMM, which leads to fast subproblem solves that only require matvecs.

**ADMM methods.** ADMM has a long history that goes back to the work of Douglas and Rachford [1956] on efficient numerical solution of the heat equation. For a complete historical discussion, see the survey Boyd et al. [2011] or the book Ryu and Yin [2022]. The popularity of ADMM in machine learning originates with the survey Boyd et al. [2011], which demonstrated ADMM could be applied successfully to various machine learning problems. Since then, there have been numerous works, Ouyang et al. [2013, 2015], Deng and Yin [2016], Zhao et al. [2022], that have developed variants of ADMM which aim to tackle the large problem instances that are ubiquitous in the era of big data.

**Randomized Numerical Linear Algebra.** Randomized numerical linear algebra (RandNLA) is a field that combines randomization with fundamental numerical linear algebra primitives to speed up computation dramatically Mahoney et al. [2011], Woodruff et al. [2014], Martinsson and Tropp [2020]. RandNLA has been successful in accelerating a variety of important computational problems, ranging from low-rank approximation Tropp et al. [2017, 2019], solving linear systems Avron et al. [2010], Meng et al. [2014], Lacotte and Pilanci [2020], Frangella et al. [2023a], to convex optimization Pilanci and Wainwright [2017], Derezinski et al. [2021], Zhao et al. [2022]. The most relevant work to our setting is Frangella et al. [2023a], which introduced Nyström PCG, the algorithm CRONOS uses to efficiently solve the $u$-subproblem (5). We are the first work to apply Nyström preconditioning in the context of convex neural networks. By leveraging this with the structure of (4) and JAX based GPU acceleration, CRONOS is able to be both fast and highly scalable.

# B   Additional algorithms

## B.1   Randomized Nyström approximation and Nyström PCG

In this section, we give the algorithms from Frangella et al. [2023a] for the randomized Nyström approximation and Nyström PCG. Algorithm 3 constructs a low-rank approximation $\hat{H} = U\hat{\Lambda}U^T$ to the input matrix $H$. $\hat{H}$ is then used to form the Nyström preconditioner $P$, which (along with its inverse) is given by:

$$P = \frac{1}{\hat{\lambda}_r + \mu}U(\hat{\Lambda} + \mu I)U^T + (I - UU^T),$$
$$P^{-1} = (\hat{\lambda}_r + \mu)U(\hat{\Lambda} + \mu I)^{-1}U^T + (I - UU^T).$$

---

**Algorithm 3** Randomized Nyström Approximation

---

**Require:** psd matrix $H \in \mathbb{S}_d^+(\mathbb{R})$, sketch size $s$

$\Omega = \text{randn}(d, s)$            $\triangleright$ Gaussian test matrix

$\Omega = \text{qr}(\Omega, 0)$            $\triangleright$ thin QR decomposition

$Y = H\Omega$            $\triangleright$ $s$ matvecs with $H$

$\nu = \text{eps}(\text{norm}(Y, 2))$            $\triangleright$ compute shift

$Y_\nu = Y + \nu\Omega$            $\triangleright$ add shift for stability

$C = \text{chol}(\Omega^T Y_\nu)$            $\triangleright$ Cholesky decomposition

$B = Y_\nu / C$            $\triangleright$ triangular solve

$[U, \Sigma, \sim] = \text{svd}(B, 0)$            $\triangleright$ thin SVD

$\hat{\Lambda} = \max\{0, \Sigma^2 - \nu I\}$ $\triangleright$ remove shift, compute eigs **return** Nyström approximation $\hat{H} = U\hat{\Lambda}U^T$

---

In CRONOS, we set $\mu = 1$, as this corresponds to the value of the regularization parameter in the linear system arising from (5). The most important thing to note about Algorithm 3, is that if the time to compute a matrix-vector product with $H$ is $T_{\text{mv}}$, the asympotic cost is $\mathcal{O}(T_{\text{mv}} r)$ Zhao et al. [2022]. The preconditioner is then deployed with Nyström PCG (Algorithm 4) at each iteration to efficiently solve (5).

---

**Algorithm 4** Nyström PCG

---

**Require:** psd matrix $H$, righthand side $r$, initial guess $x_0$, regularization parameter $\rho$, sketch size $s$, tolerance $\varepsilon$

$[U, \hat{\Lambda}] = \text{RandomizedNyströmApproximation}(H, s)$

$w_0 = r - (H + \rho I)x_0$

$y_0 = P^{-1}w_0$

$p_0 = y_0$

**while** $\|w\|_2 > \varepsilon$ **do**

    $v = (H + \rho I)p_0$

    $\alpha = (w_0^T y_0)/(p_0^T v)$

    $x = x_0 + \alpha p_0$

    $w = w_0 - \alpha v$

    $y = P^{-1}w$

    $\beta = (w^T y)/(w_0^T y_0)$

    $x_0 \leftarrow x, w_0 \leftarrow w, p_0 \leftarrow y + \beta p_0, y_0 \leftarrow y$

**end while**

**return** approximate solution $\hat{x}$

---

### B.1.1 Setting hyperparameters in Nyström PCG

Nyström PCG has two parameters: the rank $r$ and the tolerance $\varepsilon$. The tolerance is easy to set — any summable sequence suffices to ensure CRONOS will converge. Thus, we recommend setting the PCG tolerance to $\varepsilon_k = k^{-1.2}$ at each iteration. This is consistent with popular ADMM solvers such as SCS, which uses the same sequence in its CG solver [Odonoghue et al., 2016].

The rank is also not difficult to set. In our experiments, a rank of $r = 20$ worked well uniformly. So, we recommend this as a default value. However, if users wish to make the rank $r$ close to the effective dimension (see Appendix C), they can use the adaptive algorithm proposed in Frangella et al. [2023a] to select the rank. This approach iteratively builds the preconditioner and is guaranteed to terminate when $r$ is on the order of the effective dimension with high probability. This guarantees that Nyström PCG will converge to tolerance $\varepsilon$ in $\mathcal{O}\left(\log(1/\varepsilon)\right)$ iterations (see Appendix C).

### B.2 CRONOS-AM

We present pseudo-code for CRONOS-AM in Algorithm 5. For DAdapted-Adam we run the algorithm for 1-epoch to perform the approximate minimization.

---

**Algorithm 5** CRONOS-AM

---

**Require:** penalty parameter $\rho$, forcing sequence $\{\delta^k\}_{k=1}^{\infty}$, rank parameter $r$
  **repeat**
      Approximately minimize (7) with respect to $(\boldsymbol{u}, \boldsymbol{v}, \boldsymbol{s})$ using CRONOS to obtain
$(\boldsymbol{u}^{k+1}, \boldsymbol{v}^{k+1}, \boldsymbol{s}^{k+1})$                        ▷ Run CRONOS for 5 iterations
      Approximately minimize (7) with respect to $w_{1:L-2}$ using DAdapted-Adam to obtain $w_{1:L-2}^{k+1}$.
▷ Run DAdapted-Adam for 1 epoch
  **until** convergence

---

## C Proofs of main results

### C.1 Proof of Proposition 6.1

*Proof.* Recall by definition, that $F_i = D_i X_i$ and $G_i = (2D_i - I)X$, where $D_i \in \mathbb{R}^{n \times n}$ is a diagonal matrix where $D_{ii} \in \{0, 1\}$. Consequently,

$$D_i^2 \preceq I \text{ and } (2D_i - I)^2 \preceq I.$$

Thus, by the conjugation rule, we conclude:

$$X^T D_i^2 X \preceq X^T X \text{ and } X^T (2D_i - I)^2 X \preceq X^T X.$$

Recalling $A \preceq B$ implies $\lambda_j(A) \leq \lambda_j(B)$, and that $F_i^T F_i = X^T D_i^2 X$ and $G_i^T G_i = X^T(2D_i - I)^2 X$, the desired claim follows from the preceding display. $\qquad\square$

### C.2 Proof of fast $u$-update using Nyström PCG

The key quantity in establishing fast convergence of Nyström PCG is to select the rank to be on the level of the *effective dimension* of the matrix defining the linear system Frangella et al. [2023a]. Given $\mu > 0$ and a symmetric positive definite matrix $H$, the effective dimension is given by:

$$d_{\text{eff}}^{\mu}(H) = \text{trace}\left(H(H + \mu I)^{-1}\right).$$

Roughly speaking, $d_{\text{eff}}^{\mu}(H)$ gives a smoothed count of the eigenvalues of $H$ larger than the parameter $\mu > 0$. We now recall the following result which bounds $d_{\text{eff}}^{\mu}(H)$ when the eigenvalues decay polynomially.

**Lemma C.1** (Effective dimension under polynomial decay.). *Let $\mu > 0, \beta > 1/2$ and let $H \in \mathbb{R}^{N \times N}$ be a symmetric positive semidefinite matrix. Suppose the eigenvalues of $H$ satisfy:*

$$\lambda_j(H) = \mathcal{O}(j^{-2\beta}), \quad \forall j \in [P].$$

*Then,*

$$d_{\text{eff}}^{\mu}(H) \leq C\mu^{-\frac{1}{2\beta}},$$

*where $C > 0$ is some constant.*

For a proof, see Section C.1 of Bach [2013]. With these preliminaries out of the way, we now commence with the proof of Proposition 6.3.

*Proof.* Let $\hat{H}$ denote the randomized Nyström approximation of $H$ output by Algorithm 3 and define $E = H - \hat{H}$. By Lemma C.1 with $\mu = 1$ and Assumption 6.2, we have that $d_{\text{eff}}^1(H) = \mathcal{O}(1)$. As $r = \mathcal{O}\left(\log\left(\frac{1}{\zeta}\right)\right) = \mathcal{O}\left(d_{\text{eff}}^1(H) + \log\left(\frac{1}{\zeta}\right)\right)$, it follows from Lemma A.7 of Zhao et al. [2022], that

$$\mathbb{P}\left(\|E\| \leq 2\right) \geq 1 - \zeta.$$

We also obtain from Lemma 5.4 of Frangella et al. [2023a] that $\lambda_r(\hat{H}) \leq 1$. Thus, applying Proposition 4.5 of Frangella et al. [2023a] gives:

$$\kappa\left(P^{-1/2}(H + I)P^{-1/2}\right) \leq 1 + \lambda_r(\hat{H}) + \|E\| \leq 4.$$

Moreover, from the convergence analysis for CG, we know that Golub and Van Loan [2013]

$$\|\boldsymbol{u}^{k+1} - \tilde{\boldsymbol{u}}^{k+1}\|_{H+I} \le 2 \left( \frac{\sqrt{\kappa(P^{-1/2}(H+I)P^{-1/2})} - 1}{\sqrt{\kappa(P^{-1/2}(H+I)P^{-1/2})} + 1} \right)^t \|\tilde{\boldsymbol{u}}^{k+1}\|_{H+I}. \tag{8}$$

Now,

$$\|(H+I)\boldsymbol{u}^{k+1} - b\| = \|(H+I)(\boldsymbol{u}^{k+1} - \tilde{\boldsymbol{u}}^{k+1})\| \le \sqrt{\lambda_1(H+I)}\|\boldsymbol{u}^{k+1} - \tilde{\boldsymbol{u}}^{k+1}\|_{H+I},$$

Hence, by the last display with (8), we reach

$$\|(H+I)\boldsymbol{u}^{k+1} - b\| \le \sqrt{\lambda_1(H+I)}\|\tilde{\boldsymbol{u}}^{k+1}\|_{H+I} (1/3)^t,$$

The desired claim immediately follows from the last display. $\qquad \square$

### C.3 Proof of Theorem 6.4

CRONOS convergence is a consequence of the following theorem, which is a simplified version of Theorem 1 of Frangella et al. [2023b] specialized to the setting of Algorithm 2.

**Theorem C.2** (Simplified Theorem 1, Frangella et al. [2023b])**.** *Let $p^\star$ denote the optimum of* (4)*. For each $K \ge 1$, denote $\bar{\boldsymbol{u}}^{K+1} = \frac{1}{K}\sum_{k=2}^{t+1} \boldsymbol{u}^k$, $\bar{\boldsymbol{v}}^{K+1} = \frac{1}{K}\sum_{k=2}^{K+1} \boldsymbol{v}^k$, and $\bar{\boldsymbol{s}}^{K+1} = \frac{1}{K}\sum_{k=2}^{K+1} \boldsymbol{s}^k$ ,where $\{\tilde{x}^k\}_{k\ge1}$ and $\{\tilde{z}^k\}_{k\ge1}$ are the iterates produced by Algorithm 2 with forcing sequence $\{\delta^k\}_{k\ge1}$ with $\boldsymbol{u}^1 = 0, \boldsymbol{v}^1 = 0, \boldsymbol{s}^1 = 0, \lambda^1 = 0$, and $\nu^1 = 0$. Let $\tilde{\boldsymbol{u}}^{k+1}$ denote the exact solution of* (5) *at iteration $k$. Suppose that $\|\boldsymbol{u}^{k+1} - \tilde{\boldsymbol{u}}^{k+1}\| \le \delta^k$ for all $k \in [K]$. Then, the suboptimality gap satisfies*

$$\ell(F\bar{\boldsymbol{u}}^{K+1}, y) + \beta\|\bar{\boldsymbol{v}}^{K+1}\|_{2,1} + \mathbb{1}(\bar{\boldsymbol{s}}^{K+1} \ge 0) - p^\star = \mathcal{O}(1/K).$$

*Furthermore, the feasibility gap satisfies*

$$\left\| \begin{bmatrix} I_{2dP} \\ G \end{bmatrix} \bar{\boldsymbol{u}}^{K+1} - \begin{bmatrix} \bar{\boldsymbol{v}}^{K+1} \\ \bar{\boldsymbol{s}}^{K+1} \end{bmatrix} \right\| = \mathcal{O}\left(\frac{1}{K}\right).$$

*Consequently, after $\mathcal{O}(1/\epsilon)$ iterations,*

$$\ell(F\bar{\boldsymbol{u}}^{K+1}, y) + \beta\|\bar{\boldsymbol{v}}^{K+1}\|_{2,1} + \mathbb{1}(\bar{\boldsymbol{s}}^{K+1} \ge 0) - p^\star \le \epsilon, \text{ and } \left\| \begin{bmatrix} I_{2dP} \\ G \end{bmatrix} \bar{\boldsymbol{u}}^{K+1} - \begin{bmatrix} \bar{\boldsymbol{v}}^{K+1} \\ \bar{\boldsymbol{s}}^{K+1} \end{bmatrix} \right\| \le \epsilon.$$

*Proof.* From Theorem C.2, it suffices to show that for all $k \in [K]$ that

$$\|\boldsymbol{u}^{k+1} - \tilde{\boldsymbol{u}}^{k+1}\| \le \delta^k.$$

To this end, observe as $r = \mathcal{O}\left(\log\left(\frac{1}{\zeta}\right)\right)$ and Nyström PCG at the $k$th iteration runs for $\mathcal{O}\left(\log\left(\frac{1}{\delta^k}\right)\right)$, it follows from Proposition 6.3, that with probability $1 - \zeta$ for all $k \in [K]$ the output $\boldsymbol{u}^{k+1}$ of Nyström PCG satisfies:

$$\|(H+I)\boldsymbol{u}^{k+1} - b^k\| \le \delta^k, \quad \forall k \in [K].$$

Consequently, with probability at least $1 - \zeta$:

$$\|\boldsymbol{u}^{k+1} - \tilde{\boldsymbol{u}}^{k+1}\| \le \|\boldsymbol{u}^{k+1} - \tilde{\boldsymbol{u}}^{k+1}\|_{H+I} = \|(H+I)(\boldsymbol{u}^{k+1} - \tilde{\boldsymbol{u}}^{k+1})\|_{(H+I)^{-1}}$$
$$= \|(H+I)\boldsymbol{u}^{k+1} - b^k\|_{(H+I)^{-1}} \le \|(H+I)\boldsymbol{u}^{k+1} - b^k\| \le \delta^k.$$

Thus, we can invoke Theorem C.2 to conclude the proof of the first statement.

We now prove the second statement. Recall that the cost of multiplying a vector by $H$ costs $\mathcal{O}\left(ndP^2\right)$. The cost of CRONOS is dominated by two parts: (1) the cost to construct the Nyström preconditioner, and (2) the cost to solve (5) at each iteration. All other operations cost $\mathcal{O}\left((n+d)P\right)$ or less. For a rank $r > 0$, the cost of forming the Nyström approximation is $\mathcal{O}(ndP^2r)$ (see Appendix B.1). In Theorem 6.4, $r = \mathcal{O}\left(\log\left(\frac{1}{\zeta}\right)\right)$, so the total cost of constructing the preconditioner is $\tilde{\mathcal{O}}(ndP^2)$. By Proposition 6.3, with probability at least $1 - \zeta$, the total cost of solving (5) for $K$ iterations is given by:

$$\sum_{j=1}^{K} \mathcal{O}\left(ndP^2 \log\left(\frac{1}{\delta^k}\right)\right) \le K\mathcal{O}\left(ndP^2 \log\left(\frac{1}{\delta^K}\right)\right) = \tilde{\mathcal{O}}\left(\frac{ndP^2}{\epsilon}\right),$$

where the last equality follows as $K = \mathcal{O}\left(1/\epsilon\right)$. Thus, with probability at least $1 - \zeta$ the total complexity of Algorithm 2 is

$$\mathcal{C}_{\text{CRONOS}} = \tilde{\mathcal{O}}\left(\frac{ndP^2}{\epsilon}\right).$$

$\square$

## D  When does the optimal value of the convex reformulation coincide with the optimal value of the original objective?

An important practical question is when does the optimal value of the convex program Eq. (3) coincide with the optimal value of the non-convex problem Eq. (2)? Pilanci and Ergen [2020] established that the optimal value of the full convex reformulation agrees with that of Eq. (2), but this is computationally intractable. It is not apparent apriori that the optimal value of the subsampled convex program (3) will agree with the optimal value of Eq. (2).

Mishkin et al. [2022] has provided an answer to when the solutions of Eq. (3) and Eq. (2) agree. Namely, they show the optimal values agree under the following reasonable conditions: (1) the number of neurons $m$ in (2) must exceed the number of non-zeros in the optimal weights of the convex model, and (2) the sampled ReLU patterns in (3) must contain the ReLU patterns in the optimal solution to (2). While this result comforts us that using the subsampled convex model is principled, it is non-quantitative, and its conditions are difficult to verify.

Fortunately, the recent work of Kim and Pilanci [2024] provides a more quantitative bound on the gap between the optimal values of (3) and (2). Let $p^\star_{\text{MLP-NCVX}}$ denote the optimal value of (2), and $p^\star_{\text{MLP-CVX}}$ denote the optimal value of (3). Theorem 2.1 in Kim and Pilanci [2024] establishes that if $m = \Omega\left(\log(n)\right)$ and $P = m/2$, then with high probability:

$$p^\star_{\text{MLP-NCVX}} \leq p^\star_{\text{MLP-CVX}} \leq C\sqrt{\log(2n)}p^\star_{\text{MLP-NCVX}},$$

where $C > 0$ is some absolute constant. This result shows that for $P = \Omega(\log(n))$, $p^\star_{\text{MLP-CVX}}$ deviates from $p^\star_{\text{MLP-NCVX}}$ by a modest logarithmic factor. Moreover, as two-layer ReLU networks are universal approximators, it is often the case that the network is capable of perfectly fitting the training data, in which case $p^\star_{\text{MLP-CVX}} = p^\star_{\text{MLP-NCVX}} = 0$.

Combining the preceding result with Theorem 6.4, we see that CRONOS will deliver a solution with comparable objective value to $p^\star_{\text{MLP-NCVX}}$ in polynomial time. When $p^\star_{\text{MLP-NCVX}} = 0$ or the conditions of Mishkin et al. [2022] hold, CRONOS can produce a solution arbitrarily close to $p^\star_{\text{MLP-NCVX}}$ in polynomial time.

## E  CRONOS for general smooth losses

This section shows how CRONOS can be extended to general smooth losses beyond the least-squares loss.

**Beyond least-squares:** The NysADMM framework of Zhao et al. [2022] applies to losses beyond the least-squares loss. Instead of solving (5), NysADMM replaces $\ell(Fu)$ with a (regularized) second-order Taylor expansion about the current iterate $u^k$, to obtain the subproblem.

$$\begin{aligned}
\text{argmin}_{\boldsymbol{u}}\{&\ell(F\boldsymbol{u}^k) + (\boldsymbol{u} - \boldsymbol{u}^k)F^T\nabla\ell(F\boldsymbol{u}^k, y) \\
&+ \frac{1}{2}(\boldsymbol{u} - \boldsymbol{u}^k)^T(F^T\nabla^2\ell(F\boldsymbol{u}^k)F + \sigma I)(\boldsymbol{u} - \boldsymbol{u}^k) \\
&+ \frac{\rho}{2}\|\boldsymbol{u} - \boldsymbol{v}^k + \lambda^k\|_2^2 + \frac{\rho}{2}\|G\boldsymbol{u} - \boldsymbol{s}^k + \nu^k\|^2\}
\end{aligned} \tag{9}$$

The solution of (9) may be found by solving the following linear system:

$$(H^k + I)\boldsymbol{u}^{k+1} = b^k,$$

where $H^k = \frac{1}{\rho}F^T\nabla^2\ell(F\boldsymbol{u}^k)F + G^TG$. Note that the matrix defining the linear system in (9) changes from iteration to iteration. Consequently, the preconditioner should be periodically updated to maintain good performance. Other than this, there are no changes to CRONOS. CRONOS for general smooth losses is presented in Algorithm 6. Similar to Algorithm 2, the convergence of Algorithm 6 can be established using results from Frangella et al. [2023b].

---

**Algorithm 6** CRONOS for general smooth losses

---

**Require:** penalty parameter $\rho$, forcing sequence $\{\delta^k\}_{k=1}^\infty$, rank parameter $r$, preconditioner update frequency $f$
  **repeat**
    **if** $k$ is a multiple of $f$ **then**
      $[U, \hat{\Lambda}] = \text{RandNysAppx}(\frac{1}{\rho}F^T \nabla^2 \ell(F\boldsymbol{u}^k)F + G^T G, r)$ ▷ Update Nyström preconditioner every $f$ iterations
    **end if**
    Use Nyström PCG to find $\boldsymbol{u}^{k+1}$ that solves (9) within tolerance $\delta^k$
    $\boldsymbol{v}^{k+1} \leftarrow \mathbf{prox}_{\frac{\beta}{\rho}\|\cdot\|_2}(\boldsymbol{u}^{k+1} + \lambda^k)$                      ▷ Primal $\boldsymbol{v}$ update
    $\boldsymbol{s}^{k+1} \leftarrow (G\boldsymbol{u}^{k+1} + \nu)_+$                             ▷ Slack $\boldsymbol{s}$ update
    $\lambda^{k+1} \leftarrow \lambda^k + \frac{\gamma_\alpha}{\rho}(\boldsymbol{u}^{k+1} - \boldsymbol{v}^{k+1})$               ▷ Dual $\lambda$ update
    $\nu^{k+1} \leftarrow \nu^k + \frac{\gamma_\alpha}{\rho}(G\boldsymbol{u}^{k+1} - \boldsymbol{s}^{k+1})$           ▷ Dual $\nu$ update
  **until** convergence

---

# F Experimental details and additional results

In this section, we include additional details relevant to our experimental design. We also provide additional numerical results not present within the main paper.

## F.1 Hardware

All experiments were performed on an RTX-4090 GPU with 24 GB of memory and 100tFLOPS in JAX functional code. We utilize x86-64 CPU architecture with Ubuntu 22.04 OS.

## F.2 Hyperparmeters for CRONOS and CRONOS-AM

In all our experiments CRONOS (including when used as a subproblem solver in CRONOS-AM is run for 5 ADMM iterations. The number of PCG iterations varies from 5-50 depending upon the task. The rank of the Nyström preconditioner varies from $r = 10$ to $r = 20$. The value of $\rho$ is varied from 0.001 to 1 depending upon the task. For CRONOS-AM DAdapted-AdamW is always run for 1 epoch to get the non-convex weights.

In practice, $\rho$ can be selected adaptively via residual balancing. Which increases or decreases $\rho$ in such a way that ensures the primal and dual residuals (which measure the quality of the approximate solution) are balanced [Boyd et al., 2011]. This is also a common strategy employed in practice by ADMM solvers to set $\rho$ [Stellato et al., 2020, Diamandis et al., 2023]. Note however, unlike deep learning optimizers which are sensitive to the learning rate, CRONOS is guaranteed to converge for any fixed value of $\rho > 0$ [Frangella et al., 2023b]. So while a judicious choice of $\rho$ may lead to faster convergence, we are not restricted to some small band about the best $\rho$ to ensure convergence. This differs significantly from the non-convex case, where if the learning rate is not within some small band, non-convergent behavior can arise.

## F.3 Datasets

**Fashion MNIST.** The Fashion-MNIST dataset is a collection of 70,000 grayscale images, each of size 28x28 pixels, categorized into 10 classes. It includes 60,000 training samples and 10,000 test samples, and is intended as a more challenging replacement for the traditional MNIST dataset of handwritten digits. We use this dataset for the multi-class classification experiment.

**Cifar10 dataset.** This dataset involves 60,000 images of size 32x32x3 in 10 different classes, with 6,000 images per class. The dataset is divided into 50,000 training images and 10,000 test images, and is commonly used in baseline image classification tasks. Previous work in convex neural network applications have been to only reach binary classification of 2 classes on Cifar10, usually in a downsampled setting.

Table 4: Validation accuracy achieved by CRONOS on IMDB, across three different settings, with multiple varying batches averaged across seeds.

| IMDB-NFT | IMDB-FT | IMDB-DA |
|----------|---------|---------|
| 74.67%   | 94.14%  | 76.15%  |

**ImageNet Dataset.** ImageNet is a large-scale dataset with approximately 14 million images have been hand-annotated by the project to indicate what objects are pictured. Each image is padded to 512x512x3 for binary classification. We provide a downsampled version of size 171x171x3 of ImageNet in order to examine scaling abilities of all methods. Our peak validation accuracy with CRONOS is 88.72%

**Food-5k Dataset.** The Food-5K dataset is a collection of 5,000 images with binary classification label of '1 = food' and '-1 = non-food'. Each image is padded to size 256x256x3 in all experiments. We select this dataset for the intermediate scaling of size between downsampled ImageNet171 and full size ImageNet512. Interestingly, we note that CRONOS performs significantly better on larger data and harder tasks with virtually zero tuning of hyperparameters.

**IMDb Dataset.** The IMDb dataset is a set of 50,000 movie reviews from the Internet Movie Database (IMDb) for binary sentiment classification. It is evenly split into 25,000 reviews for training and 25,000 reviews for testing. Each set consists of an equal number of positive and negative reviews, where the sentiment of each review is determined by the number of stars awarded by the reviewer. It consists of only negative and positive reviews, without neutral. We perform all experiments with a batch size of 32, and pad all input sequences to length 60 with the PAD token defined as 'EOS'. Input to CRONOS is reshaped into (1500x60x768), with 1500 samples and an embedding dimension of 768. Our best accuracy on the GPT2 foundation experiments with CRONOS yields 94.14% validation accuracy.

## F.4    Details on Three NLP Experiments with GPT2

We conducted 14 NLP experiments grouped into 3 categories (no train GPT2-NFT, 1 epoch fine tuning GPT2-FT, and unsupervised GPT2-DA). In each experiment, we benchmark against varying seed and learning rate sweeps of AdamW as a baseline. We select AdamW as our baseline since it is predominantly utilized in most NLP tasks. Our results show that CRONOS outperforms tuned AdamW in both settings in terms of speed and accuracy.

**IMDB-NFT.** We initially extract the GPT2 pretrained checkpoint and then immediately follow with CRONOS for sentiment classification. Notably, this setting does not involve any training loops with standard optimizers (such as AdamW), and the foundation GPT2 model does not see any labeled sentiment data from the IMDB dataset. We limit the amount of data seen by CRONOS to only 2 batches only, in order to evaluate the efficacy of our method on low data regime settings.

**IMDB-FT.** Our second language model experiment uses the GPT2 pretrained checkpoint followed by *one epoch only* of training with the AdamW optimizer on default parameters with the standard dense linear layer head followed by CRONOS. Although the authors of BERT recommend 2-4 epochs for training in this setting, we aim to evaluate the performance of CRONOS with limited overhead to extract features for maximum efficiency. The resulting output of the dense layer is then input to CRONOS for the sentiment classification task, and demonstrates extremely promising results of approximately 94% validation accuracy after utilizing 2 batches of labeled data. This is particularly significant due to the lack of hyperparameter tuning in CRONOS, and its consistency across multiple trials of varying data batches.

**IMDC-DA.** Finally we examine the setting of training for one epoch of AdamW on *unlabeled* IMDB data initialized from the GPT2 checkpoint. This experiment was motivated by the idea of subspace alignment. Since pre-trained GPT2 has vast semantic knowledge, even minor alignment to an unlabeled dataset should be beneficial to the eventual downstream task. We are also practically motivated to explore challenging settings with unlabeled data, since this is readily prevalent in real world scenarios. We run one epoch of unlabeled IMDB data through a pre-trained GPT2 to predict the next token in the language modeling setting. This new checkpoint is then used for feature extraction and classification with our convex network and AdamW. This examines the potential of CRONOS on

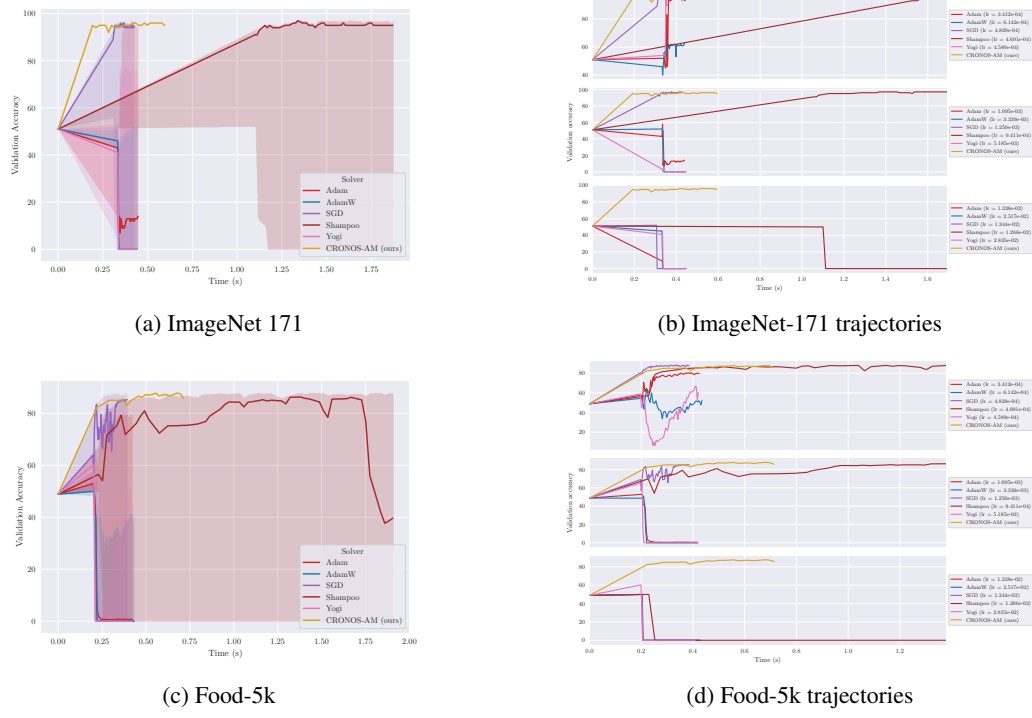

(a) ImageNet 171

(b) ImageNet-171 trajectories

(c) Food-5k

(d) Food-5k trajectories

Figure 3: Results for ImageNet-171 and Food-5k

unsupervised domain adaptation settings in future work, which aims to reduce the distribution gap between source and unlabeled target domains. We recognize that domain-specific tasks generally perform well when the feature distributions of the domains are similar. Therefore we examine the effectiveness of initializing from the broadly pretrained GPT2 model on the unlabeled IMDb dataset. Experimental results show promising directions for future work.

### F.5 Additional Results for the MLP experiment

Below in Fig. 3, we report results for various optimizers on Food-5k.

### F.6 Convolutional Neural Network Experiments

Here we test the performance of CRONOS-AM on a 4-layer CNN consisting of 2 convolutional layers with average pooling, followed by two densely connected layers. The goal of this experiment is to demonstrate the feasibility of applying CRONOS-AM to architectures aside from MLPs. Fig. 4 gives results for CIFAR-10 and ImageNet-171. On CIFAR-10, CRONOS-AM performs somewhat worse than the competition, but delivers excellent performance on ImageNet-171, where it reaches high accuracy quite quickly. These experiments validate the applicability of CRONOS-AM beyond MLPs. For future work, it would be interesting to explore the effectiveness of CRONOS-AM on other popular architectures.

### F.7 Multiclass Classification Experiment

To see how CRONOS performs in the multi-class setting, we experiment on the Fashion MNIST dataset [Xiao et al., 2017]. This yields a classification problem with 10 unique classes. We consider a two-layer ReLU MLP with 64 neurons. For comparison we also consider Adam, AdamW, SGD+Momentum, Shampoo, and Yogi. To fully exploit GPU acceleration, the batchsize for these methods is set to $10,000$. As in the main paper, we consider a random grid of five learning rates. The results for the experiment are provided in Table 5. The table shows CRONOS delivers the best performance by significant margin relative to its non-convex competitors. Moreover, we see the non-convex stochastic

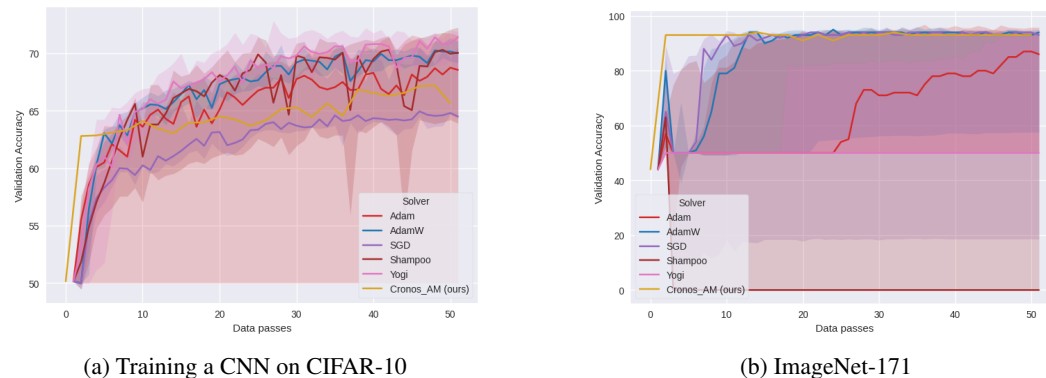

(a) Training a CNN on CIFAR-10       (b) ImageNet-171

Figure 4: Training a CNN on ImageNet-171

optimizers are very sensitive to the learning rate. Again, CRONOS does not have this issue, as it does not require a learning rate.

Table 5: Results for Fashion MNIST

| | Two-layer ReLU MLP with 64 neurons | |
| --- | --- | --- |
| Optimizer | Peak Validation Range | Best Learning Rate |
| CRONOS | **85.8**% | NA |
| Adam | $[12.24, 78, 62]\%$ | $3.52 \times 10^{-4}$ |
| AdamW | $[12.23, 78.61]\%$ | $3.52 \times 10^{-4}$ |
| SGD | $[6.63, 75.32]\%$ | $1.97 \times 10^{-2}$ |
| Shampoo | $[5.87, 78.49]\%$ | $2.00 \times 10^{-2}$ |
| Yogi | $[19.44, 71.54]\%$ | $3.52 \times 10^{-4}$ |

## F.8  Results for different random seeds

In this subsection we report results for the deep MLP experiments for each dataset for two different random seeds. Looking at the figures, it is clear the trends we observed earlier in the main paper are stable across the random seed.

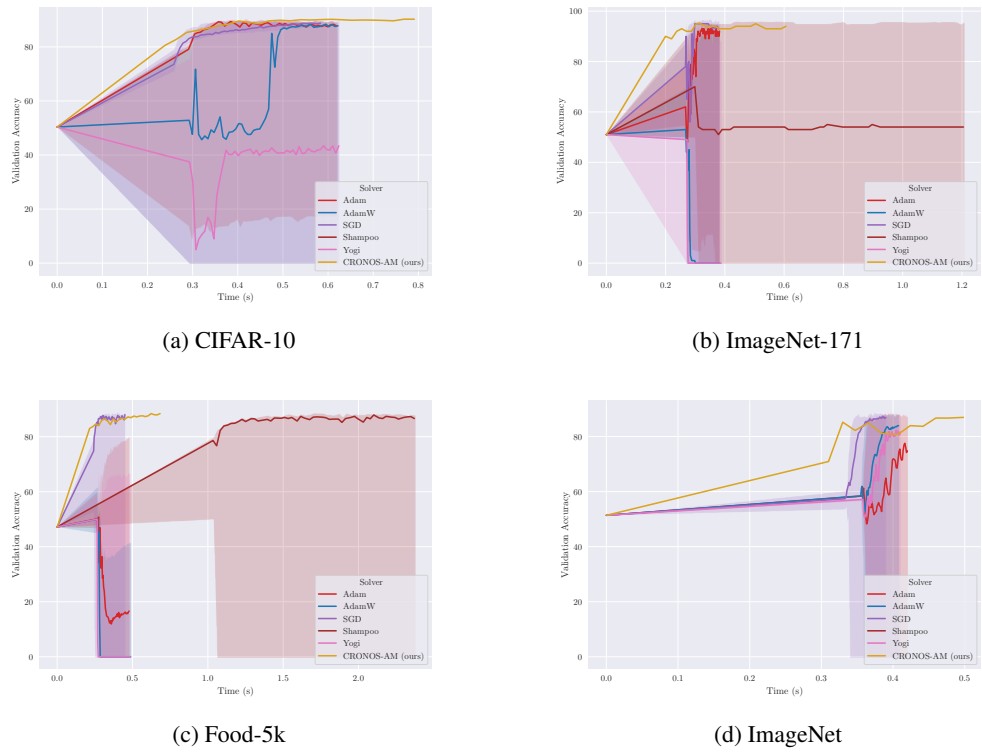

(a) CIFAR-10

(b) ImageNet-171

(c) Food-5k

(d) ImageNet

Figure 5: CRONOS-AM vs. competitors on Deep ReLU MLP (Seed 2)

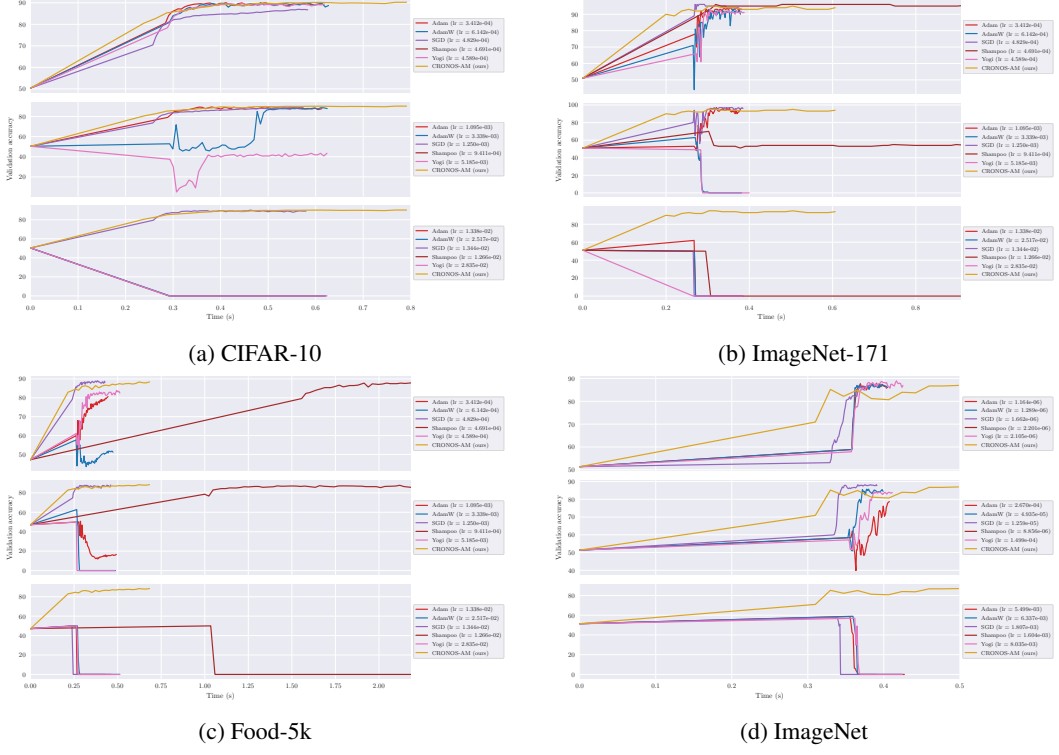

(a) CIFAR-10

(b) ImageNet-171

(c) Food-5k

(d) ImageNet

Figure 6: Learning rate trajectories for Deep ReLU MLP (Seed 2)

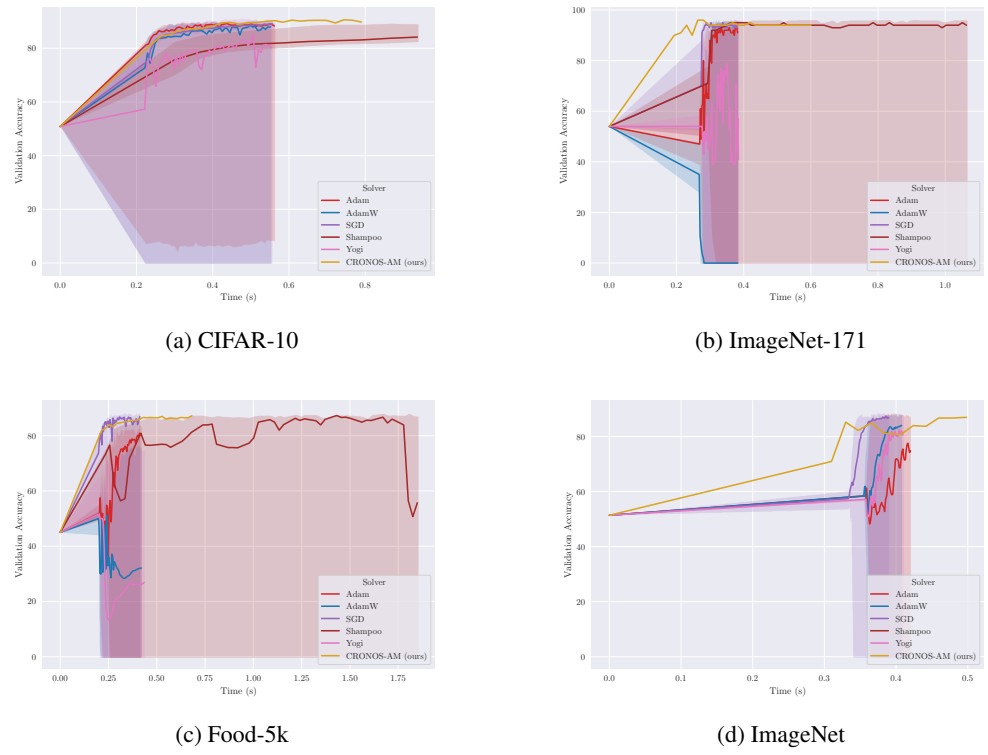

(a) CIFAR-10

(b) ImageNet-171

(c) Food-5k

(d) ImageNet

Figure 7: CRONOS-AM vs. competitors on Deep ReLU MLP (Seed 3)

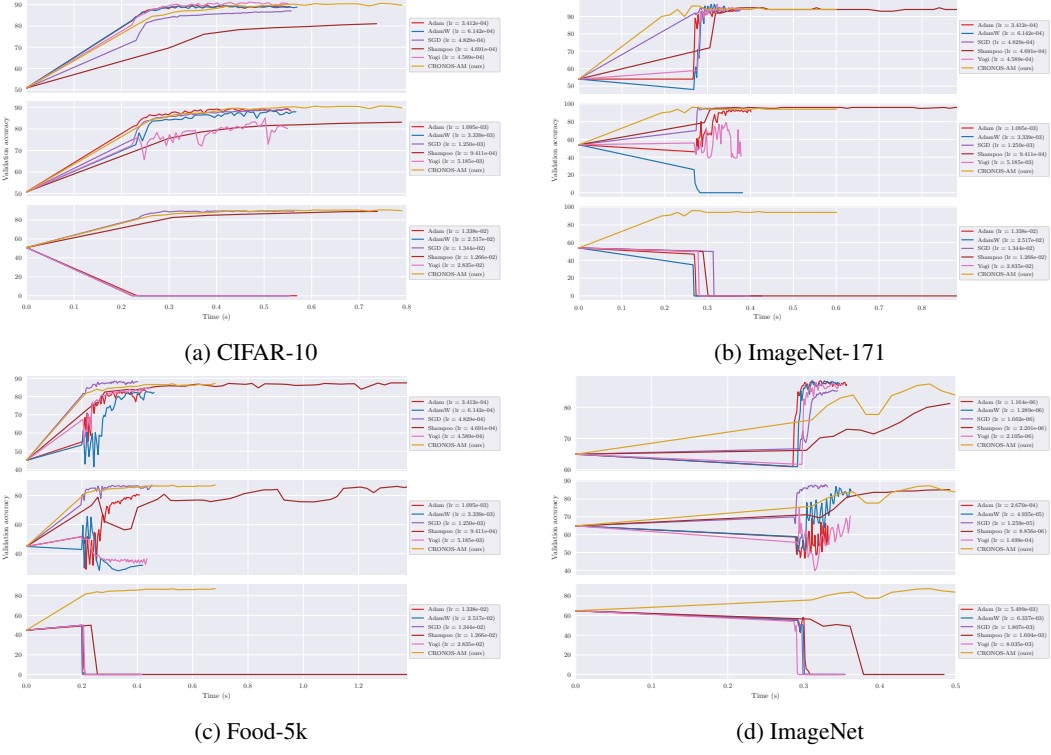

(a) CIFAR-10

(b) ImageNet-171

(c) Food-5k

(d) ImageNet

Figure 8: Learning rate trajectories for Deep ReLU MLP (Seed 3)

