# OpenReview forum: "CRONOS: Enhancing Deep Learning with Scalable GPU Accelerated Convex Neural Networks"
_NeurIPS.cc/2024/Conference — NeurIPS 2024 poster_

### Official Review · Reviewer_dmGV · 2024-07-09

**Soundness:** 3
**Presentation:** 3
**Contribution:** 2
**Rating:** 5
**Confidence:** 3

**Summary:**

- This paper builds on work on reframing the optimization of 2-layer ReLU networks as convex optimization
- The first contribution of this work is the use of the ADMM algorithm for solving the constrained optimization task from equation 4
- Noting that need of repeatedly solve the inner problem in the ADMM method, they propose using a Nystrom-preconditioned solver to accelerate the inner problem, leading to Cronos
- For deeper networks, the authors propose alternately minimizing the earlier model parameters (while freezing the final two layers) with D-adapted Adam, then optimizing the the final two layer parameters with with Cronos, leading to Cronos-AM
- They show that this method can successfully optimize deep networks, leading to the first successful application of convex neural network application for deep networks

**Strengths:**

- This work is the first work to successfully apply the convex reformulation for optimizing deep networks, albeit only the final two layers are optimized solely using this Cronos
- The theory is robust and well supported
- Compared to existing methods, Cronos does not require hyperameter tuning (such as learning rates), and appears to perform well compared to existing methods, albeit on binary classification tasks (for both vision and language)

**Weaknesses:**

- It seems that the use of D-adapted Adam for the majority of the network layers is necessary for the Cronos-AM to work. Therefore, it is unclear how important Cronos actually is
- The authors claim that an advantage of Cronos is the lack of hyperparameters, but does not compare to D-adapted optimizers in figures 1 and 2, which similarly do not have hyperparameters
- There is little to no discussion on the actual empirical runtime of the algorithm (for example, how many hours/minutes does the method take compared to existing methods?)

**Questions:**

- Would it be possible to have the results from Figure 1 and 2 in a table format for easier reading?
- The appendix contains only the performance of Cronos on Fashion-MNIST on multiclass classification. It would be useful to include the performance of existing methods for a fair comparison
- Are all other optimizers really achieving 0% accuracy on Binary ImageNet in Figure 1B? For binary classification achieving 0% would mean guessing the opposite class every since time, which seems like the classes were inverted. I am skeptical whether the plots in figure 1b are accurate.
- Could we see the performance of D-adapted methods in figure 1 and 2?
- What is the observed runtime of the algorithm?
- The authors claim that the method is appealing because of the lack of hyperparameters, yet it seems that there are a number of hyperpameters associated with the Nystrom PCG algorithm (such as the rank and the tolerance). It would useful to include results showing the effect of these hyperparameters on the method's performance.

**Limitations:**

See the limitations section. Overall, I think this is an interesting line of work and successfully applying the convex formulation of two-layer networks for deep networks is a solid accomplishment, but the papers seems to be lacking proper comparisons to baselines and also does not thoroughly discuss aspects of the proposed methods (such as sensitivity of hyperparameters). I (the reviewer) am not particularly familiar with the work on convex-neural networks, however, so it is possible that I am undervaluing the theoretical contributions of this work.

---

> ### Author Rebuttal · Authors · 2024-08-06
>
> **Weaknesses**
>
> 1. This is an excellent point that was unclear in the original submission. We have included figures for CIFAR-10 and ImageNet in the rebuttal document which include D-Adapted Adam (DAdam)  in the comparison. The difference in performance between CRONOS-AM and DAdam are visible on trajectory Figures (a) and (b). We have also summarized numerical comparisons in Tables 1 and 2 of the rebuttal document. It is evident that CRONOS-AM outperforms DAdam, which is shows the importance/value of CRONOS.
>
>  We recommend DAdam for optimizing layers $1:L-2$ since it does not require tuning. This makes CRONOS-AM easier to use in practice. By applying the convex reformulation in the final two layers CRONOS-AM obtains better performance and faster convergence to solutions of better quality than DAdam. Alternatively, any other deep learning optimizer can be used for layers $1:L-2$.
>
> We will include in the revision figures and plots similar to those in the rebuttal, to make the benefits of the convex reformulation in the last two layers clear.
>
> 2.  Please see the preceding response.
>
> 3.  Below is a table showing the runtime of CRONOS and CRONOS-AM to non-convex optimizers.
> Notably, the runtimes are comparable to standard methods, and we can conclude that the proposed methods have reasonable empirical complexity. In addition, we will include a more comprehensive table of runtimes in the revised document, summarizing NLP experiments with larger data sizes.
> ## Optimizer Runtimes (s) on CIFAR-10 and ImageNet
> | Dataset | Cronos-AM | Adam | AdamW | D-Adapted Adam| SGD | Shampoo | Yogi |
> | --------- | :---------:| :---------: | :---------: | :---------: | :---------: | :---------: | :---------: |
> | CIFAR-10 | 3.00 | 3.02 | 3.14 | 3.68 | 2.28 | 6.80 | 2.79|
> | ImageNet | 5.10 | 1.84| 3.14| 2.94|  2.48| 5.19| 1.98
>
> **Questions**
> 1. Yes, this is a great idea! We agree this helps make things more clear, and shall include tables like those presented in the rebuttal pdf.
>
> 2.  Yes, this is a good point!
>  We plan to add these results to the revision.
>
> 3. This occurred because our learning rate grid for tuning the competitors was not fine enough. We have re-run our experiments with a finer learning rate grid, and the revised plot replacing Figure 1.1b) is included in the rebuttal pdf Figures (a) and (b). The competing methods do not all fail.
>
> 4. Yes! We have added this comparison.
>     Please see the rebuttal pdf.
>
> 5.  Please see earlier response with table of runtimes, thank you. We will include a more comprehensive table of runtimes for all    experiments in the final document.
>
> 6. The reviewer raises an excellent point! Nyström PCG does have two hyperparameters.
>     Fortunately, they are very easy to set.
>     For the tolerance, one can use any summable sequence to ensure convergence.
>     In particular, the popular ADMM solver SCS from [1] uses a sequence of $1/k^{1.2}$.
>     Another common heuristic is to use the geometric mean of the ADMM primal and dual residuals. This is what OSQP [2] uses, and also was used by [3].
>     We recommend using the summable tolerance as that is what the theory requires to obtain convergence.
>     Nevertheless, past work has found that both these strategies perform very similarly [3].
>     We find this to be true in our experience as well.
>     We will include a discussion of how to set the tolerance in the revision.
>
>     The rank is also not difficult to set; in all our experiments, we found $r = 20$ works well in all our experiments.
>     This is consistent with works such as [3], which used a rank of $r = 50$ in all there experiments.
>     We also note that [4] which proposed the Nyström preconditioner also comes with an adaptive rank selection algorithm that provably selects the optimal rank.
>     Since we found it easy to select the rank, we did not use this algorithm.
>     In the revision, we will include a more thorough discussion about how to select $r$.
>     We will also mention the adaptive algorithm of [4] to select the rank.
>
> **References**
>
> [1] O’donoghue et al., Conic optimization via operator splitting and homogeneous self-dual embedding (2016)
>
> [2] Schubiger et al., GPU acceleration of ADMM for large-scale quadratic programming (2020)
>
> [3] Zhao et al., NysADMM: faster composite convex optimization via low-rank approximation (2022)
>
> [4] Frangella et al., Randomized nyström preconditioning (2023)

---

> > ### Comment · Reviewer_dmGV · 2024-08-13
> >
> > I appreciate the additional experiments and they have addressed my concerns. I will raise my score to a 5.
> >
> > Additionally, what is the runtime table measuring? Could the authors just give a table showing the total runtime of the entire training procedure? Unless I am misunderstanding, 3.00s is too fast to train a model, so the numbers in that table do not seem correct.

---

> ### Author Response · Authors · 2024-08-14
> **Thank you for your reply**
>
> We thank the reviewer for their reply and for raising their score. We are happy to hear we addressed your concerns.
>
> The table shows the total time each optimizer takes to train the model. The reviewer is correct in noting that the training time is quite fast! The main reason for this is that all optimizers use JAX's just-in-time compilation (JIT) along with GPU acceleration, making all the methods extremely fast. Additionally since these experiments consider binary classification with two classes, the number of training examples isn't massive. Therefore training for 50 epochs happens very fast when using JIT on a 4090 GPU with about 83 TFLOPS. Without Jax's JIT compilation the runtimes would not be this fast.

---

### Official Review · Reviewer_JCYR · 2024-07-12

**Soundness:** 3
**Presentation:** 3
**Contribution:** 3
**Rating:** 5
**Confidence:** 3

**Summary:**

The authors introduce a reformulation of optimizing a two-layer neural network (one neuron in the second layer) to a convex optimization problem and suggest an algorithm to solve it.
This is done via reformulating a convex program based on sampled P activation patterns from the set of all possible ReLU activation patterns. The optimization is done from the space of P convex cones.  (I have some concerns/questions here).

Then Proposition 3.1., shows how to take a step further to define it as a constrained generalized linear model that can be solved via ADMM (given by algorithm 1).
The first line of algorithm 1 requires solving a linear system (6) -- Solving (6) directly is too expensive due to large matrix sizes.
The Conjugate Gradient (CG) algorithm, needing only matrix-vector products, offers an alternative but converges slowly because of high condition numbers in typical machine learning data matrices, making CG impractically slow for repeated use.

To speed up CG convergence, they exploit the problem's approximate low-rank structure. The subproblem for v^{k+1} and s^{k+1} has a closed-form solution computable in O((n+d)P) time. The authors suggest applying the NysADMM algorithm, using Nyström Preconditioned Conjugate Gradient (NysPCG) to solve (6). NysPCG constructs a low-rank preconditioner for H+I, solving the linear system to \delta-accuracy in O(log(1/\delta)) iterations.

Then CRONOS-AM is suggested for optimizing full DNN that integrates CRONOS with alternating minimization, allowing for the training of multi-layer networks with any architecture.

The theory seems to be robust and very interesting.

**Strengths:**

1. The paper has many theoretical novelties and seems interesting from this perspective.

2. The used tricks and formulation can open new directions for research.

3. The paper aims to fill a big gap in deep learning --> theory to practice, which is a very important problem to tackle.

**Weaknesses:**

Small concerns:

1. Writing: The authors assume the readers have robust knowledge in many fields and do not provide explanations, requiring so much time to understand and read the paper.

2. The authors should be clear when defining their claims, e.g., Equation (1) does not define a classic 2-layer relu (with softmax). It is a one-hidden layer relu with a single neuron on the output layers.

Big concerns:

3. Experiments:
a. What do the authors mean by binary classification on imagenet and cifar?!
b. Figure 2 seems to be wrong --  please add also the training accuracy to see if it is simply overfitting or what is happening.
    * Adam is usually better than SGD, I encourage the authors to try the standard common parameters of ADAM.
    * On what dataset is Figure 2 reporting the results?

c. Figure 1 b is even worse, how come, known optimizers do not perform well on binary classification? and even reach 0 Accuracy

d. NLP:
   * What is the accuracy of GPT before running your algorithm on the same data, without any fine-tuning?
   * I did not understand the baselines in the nlp task, do you use coronos in all of them,? if yes, why you did not compare your algorithm to other optimizers on the NLP task just simply to finetune or train one layer as a classifier?

I don't feel that the experiments are convincing or even correct; I don't think that only cronos can optimize a binary classification problem in IMagenet and the other famous known algorithms fail.

**Questions:**

* Consider changing the notation of relu it is so confusing specifically that the next character is multiplied and not added.

* The authors define W^(1) and then use W_1 ?

* I am missing the connection from (2) to (3):
     A. Why sampling P activation patterns is enough?
     B. Is it simple by sampling?

* Define ||v||2,1 and provide an intuition on (4)

* what are the reasonable conditions that (3) still has the same optimal solution as (2)?

**Limitations:**

See above

---

> ### Author Rebuttal · Authors · 2024-08-06
>
> **Weaknesses**
>
> 1. Thank you for raising the point! We understand that the paper covers a wide range and utilizes ideas from diverse subjects such as deep learning, convex optimization, randomized numerical linear algebra, and high-performance computing. Unfortunately given the page constraints, we couldn't provide as much background information in the main paper as we would have liked. We are happy to include more background in the supplement of the revision with crossrefs in the main paper. If the reviewer has any thoughts on what would background info would be helpful, please let us know!
>
> 2. Equation (1) does define a 2-layer ReLU networkin the regression setting, see [1, 2] .
>    The more familiar softmax case mentioned by the reviewer can also be covered by the convex reformulation, see [1, 3] for further details.
>
> 3a. By binary classification, we mean we look at two classes of CIFAR-10 and ImageNet and perform binary classification.
>         This is consistent with prior literature on convex neural networks such as [3, 4}.
>
> 3b. This occurred since the initial learning rate grid we used wasn't fine enough.
>       Our original grid included Adam's default learning rate of $10^{-3}$.
>        However this value doesn't work well in the architecture considered for these two classes of ImageNet.
>        The new plot is included in the rebuttal pdf Figures (a) and (b) with a finer grid.
>        Now, the competing methods don't all fail.
>
> 3c.  Figure 2 shows the performance of Adam and SGD across different learning rates on CIFAR-10.
>        As per the previous response, the issue with figure 1b) has been revised with the finer grid in the rebuttal pdf. We have also revised the captions of our figures to indicate the dataset clearly.
>
> 3d.  Thank you for raising this question! Since we utilize the features extracted from GPT2, the immediate output does not readily report a classification accuracy. We do perform experiments across 3 configurations of feature extraction with GPT2 (without fine-tuning) and benchmark against various trajectories of AdamW as a baseline. These three configurations are:
>
> *  Features extracted from pre-trained GPT2 followed by classification immediately with No Fine-Tuning (IMDb-NFT).
> *  Features extracted then followed by a mean-pool layer before classification.
> * Features extracted then followed by an attention-pooled layer followed by classification.
>
> In all three settings above we benchmark against AdamW as a baseline, since this seems the most predominate in NLP applications. An example of these results have been included in the rebuttal pdf (Figures c -f), and we will definitely add more details to clarify experiment settings in the revision.
>
> **Questions**
> 1.   Sure! We'll change this in the revision to avoid any confusion.
> 2. Thanks for catching this!  We will fix it in the revision.
> 3. Eq (2) is the standard non-convex training problem.
>    [2] proved that this admits an equivalent convex reformulation based on sampling patterns from the set $\mathcal D_X$. Although this set is discrete, its cardinality grows as $\mathcal O (r(n/r)^r)$, where $r$ is the rank of the data matrix.
> This is extremely large unless the rank is small, so sampling all the patterns from this set is computationally infeasible.
>
>  Eq (3) uses the same objective as the reformulation in [2], but only works with a subset of $P$ patterns from $\mathcal D_X$. This result yields a convex optimization problem that is a computationally tractable alternative to Eq (2).
>
> This is a great question! Theorem 2.1 in [3] gives precise conditions under which the optimal values of Eq (1) and Eq (2) coincide.
> But a simpler, more quantitative explanation is provided by Theorem 2.1 in the recent paper [5].
> This theorem shows that as long as $P = \mathcal O\left(\log(n)\right)$, where $n$ is the number of data points, then
>
> Optimal value of Eq 2 $\leq \mathcal O(\sqrt{\log(n)}) \times$ Optimal value of Eq 1.
>
> Evidently the optimal value of the subsampled convex reformulation Eq (2) is upper bounded by an extremely modest multiple of the optimal value of the non-convex problem Eq (1).
> In this work, we only used $P = 10$ throughout all experiments, which was sufficient to obtain good results.
> In the revision, we include a more careful discussion of these points.
>
> As per your question on whether sampling is efficient, the answer is yes!
>  We can obtain an element of $\mathcal D_X$ by sampling a standard normal random vector $g \in \mathbb R^{d}$ and computing $\textup{ReLU}(Xg)$, which costs $\mathcal O(nd)$. Sampling $P$ patterns costs $\mathcal O(ndP)$.
>
> 4. Thank you for catching this!
>     We will make sure it is defined in the revision.
>
>     Sure, we can give intuition!
>     The $\mathcal K_i$ in (3) are defined by linear inequality constraints.
>     By adding non-negative slack variables, these can be turned into linear equality constraints.
>     Then, organizing all the variables into the appropriate vectors $\mathbf u, \mathbf v,$ and $\mathbf s$,
>     and compactly writing down the linear equality constraints, one arrives at (4).
>     As we note in the paper, this formulation was previously derived in [4], but we are happy to include a derivation in the supplement to help the paper be more self-contained.
>
> 5.  Please see the response to question 3.
>
> **References**
>
> [1] Du et al., Gradient Descent Provably Optimizes Over-parameterized Neural Networks (2018)
>
> [2] Pilanci and Ergen, Neural Networks are Convex Regularizers: Exact Polynomial-time Convex Optimization Formulations for Two-layer Networks (2020)
>
> [3] Mishkin et al., Fast Convex Optimization for Two-Layer ReLU Networks: Equivalent Model Classes and Cone Decompositions, (2022)
>
> [4] Bai et al., Efficient global optimization of two-layer relu networks: Quadratic-time algorithms and adversarial training (2023)
>
> [5] Kim and Pilanci, Convex Relaxations of ReLU Neural Networks Approximate Global Optima in Polynomial Time (2024)

---

> > ### Comment · Reviewer_JCYR · 2024-08-10
> > **Post rebuttal**
> >
> > Thank you for the detailed response.
> > It addressed most of my comments.
> >
> > I will increase the score as I see the main contributions of this paper are the theoretical ones. However, I still think that the experiments are limited (even if prior work does that).

---

> > > ### Author Response · Authors · 2024-08-11
> > > **Thank you for your reply**
> > >
> > > Thank you for your response and for raising your score. We are happy to hear that our reply addressed your main concerns.

---

### Official Review · Reviewer_inX1 · 2024-07-19

**Soundness:** 2
**Presentation:** 3
**Contribution:** 2
**Rating:** 5
**Confidence:** 3

**Summary:**

This work investigated the optimization of convex reformulated neural networks. The authors proposed to solve the sub-sampled convex optimization problem (for a two-layer ReLU network) with operator splitting (using ADMM), and used conjugate gradient method with Nystrom preconditioning to solve one subproblem in ADMM iterations. The proposed Nystrom preconditioning utilizes the low rank nature of data. The authors further proposed to extend the method to multi-layer networks through alternative minimization. The authors provided an efficient GPU implementation of the proposed algorithm using JAX and just-in-time (JIT) compilation.

The authors theoretically (1) verified that the subproblem in ADMM inherits the low-rank structure of data and thus can enjoy fast convergence through Nystrom preconditioning; (2) proved the fast convergence of the ADMM subproblem with conjugate gradient method; (3) and proved the convergence of the overall algorithm to the optimal solution of the sub-sampled convex optimization. The authors also showed that the proposed algorithm has a lower computation complexity compared to a previous work that directly applies ADMM, and thus is more scalable to larger datasets such as ImageNet.

The authors conducted extensive empirical experiments and claimed to be the first that successfully scale convex reformulated neural network optimization to large data such as ImageNet and large language modeling tasks with GPT2 architecture.

**Strengths:**

1. As the authors claimed, this work is the first practically useful algorithm for training neural networks in their convex reformulation on large and complex datasets. And the authors verfied this with promising empirical results.

2. The approximate solver with conjugate gradient method plus Nystrom preconditioning looks a practical and promising approach and the authors did provide JAX implementation that will be useful for the following research.

**Weaknesses:**

1. The description of how the hyperparameter $P$ is selected is missing and the authors did not provide any discussion on this hyperparameter, but I suppose it is important for the following reasons.
    * Although the authors proved the convergence of CRONOS to the global minimum of (4), the closeness of the global minimum of (4) to that of (3) depends on the effective sampling of $D_X$.
    * According to Mishkin et al. [2022], the global minima of the two problems coincide if the hidden dimension is large enough AND the optimal $D_i$ are contained in the sampled subset.
    * Bai et al. [2023] also show that to make the optimal objective of (4) effectively small, we need $P$ to increase linearly as the number of data increases.
    * The selection of $P$ directly influences the computation cost as the authors analyzed in Section 6.4.

However, as far as I observed, the authors did not mention the selection of $P$ in practice, while claiming the proposed CRONOS is almost hyperparameter tuning free.

2. Following the first point, the authors did not mention the actual training time comparison of the proposed method and normal deep learning optimization methods.

**Questions:**

1. How were the experiments on Imagenet actually conducted?
    * Why were they binary classification tasks as described in Line 599?
    * Did the authors also use a type of stochastic opitmization with CRONOS-AM so that it can be applied to more than 14M training images?

2. In IMDB experiments, the authors did not compare CRONOS with existing optimizers as they did in Figure 2, while they did use 1 epoch of AdamW training before CRONOS optimization as they described in Line 622-624. Can the authors clarify the experiment settings more clearly and explain how can we decouple the effect of AdamW and CRONOS optimization?

3. Notation $P$ was abused. It was used to denote the number of $D_i$ samplings in Section 3 and later the pre-conditioning matrix in Section 4.

4. In line 91, it should be "provided $m \geq m^*$, for some $m^* \leq n + 1$.

---

> ### Author Rebuttal · Authors · 2024-08-06
>
> **Weaknesses**
>
> 1.) This is an excellent point! It is important to have guidance on how to select the hyperparameter $P$  in practice.
>  Fortunately, large $P$ is not necessary to ensure the optimal values are comparable.
>  The recent work [1] (see Theorem 2.1) on convex neural networks has shown that as long as the number of neurons satisfies $\mathcal O\left(
>  \log(n)\right)$, the optimum of the convex reformulation will deviate from the optimum of the original two-layer ReLU problem by a multiplicative factor of at most $\mathcal O\left(\sqrt{\log(n)}\right)$.
>  This is an extremely modest multiple.
>  Therefore from a practical standpoint, only a modest number of neurons is needed to ensure good performance.
>  This agrees with practice, as in the experiments in the main paper we have only used $P=10$ across all experiments, but we obtain comparable or better performance relative to non-convex optimizers.
>
>  In the revision we will be sure to mention this point.
>  We will also include a more formal discussion in the supplement about when the optimal solutions actually coincide, which occurs when the conditions of Mishkin et al. (2022) mentioned by the reviewer hold.
>
> 2.) Below is a table showing the runtime of CRONOS and CRONOS-AM to non-convex optimizers.
> Notably, the runtimes are comparable to standard methods, and we can conclude that the proposed methods have reasonable empirical complexity. In addition, we will include a more comprehensive table of runtimes in the revised document, summarizing NLP experiments with larger data sizes.
> ## Optimizer Runtimes (s) on CIFAR-10 and ImageNet
> | Dataset | Cronos-AM | Adam | AdamW | D-Adapted Adam| SGD | Shampoo | Yogi |
> | --------- | :---------:| :---------: | :---------: | :---------: | :---------: | :---------: | :---------: |
> | CIFAR-10 | 3.00 | 3.02 | 3.14 | 3.68 | 2.28 | 6.80 | 2.79|
> | ImageNet | 5.10 | 1.84| 3.14| 2.94|  2.48| 5.19| 1.98
>
> **Questions**
>
> 1.)     We randomly selected two classes on ImageNet and performed binary classification. As the reviewer suggests, to be able to handle all of ImageNet would require replacing CRONOS in CRONOS-AM with a stochastic variant. We believe a stochastic variant of CRONOS-AM is the next step in advancing the practical deployment of convex NNs. It is an exciting direction for future work that we will mention in the revision.
>
> 2.) Thank you for raising this concern! We have thoroughly added more experimental results, numerical tables, and setting details to the revised document for clarification. We conducted 14 NLP experiments grouped into 3 categories (no train GPT2-NFT, 1 epoch fine tuning GPT2-FT, and unsupervised GPT2-DA). In each case we benchmark against seed and learning rate sweeps of AdamW on the binary classification task. Please see Figures (c) - (f) in the rebuttal pdf for a sample of run time plots. CRONOS is able to outperform AdamW in both peak validation and in the time needed to reach it. In the GPT2-NFT (No Fine Tune) setting, the effects of CRONOS and AdamW are examined separately. Features extracted from a pre-trained GPT2 are given directly to each optimizer for individual classification. We experiment with 3 scenarios in the no fine-tune setting: immediate classification, mean-pool then classification, and attention-pooled then classification. In each case CRONOS achieves higher test accuracy than AdamW (with extensive grid search).
>
>
> 3, 4.) Thank you for catching these! We will fix them in the revision.
>
> **References**
>
> [1] Kim and Pilanci, Convex Relaxations of ReLU Neural Networks Approximate Global Optima in Polynomial Time (2024)

---

### Official Review · Reviewer_iBsY · 2024-07-23

**Soundness:** 3
**Presentation:** 3
**Contribution:** 3
**Rating:** 7
**Confidence:** 3

**Summary:**

In the paper, two neural network optimization methods, CRONOS and CRONOS-AM, are proposed. The authors provide theoretical complexity analysis and convergence proof of the algorithms, and perform experiments on large datasets of image and language tasks to verify the effectiveness of the algorithm. As can be seen from the experimental results, CRONOS provides better or comparable performance compared to optimizers such as SGD, Adam, and AdamW, but hardly any tuning of hyperparameters. This is the first time that convex networks have been applied to large datasets with competitive performance, which has great significance.

**Strengths:**

1. Two neural network optimization methods, CRONOS and CRONOS-AM, are proposed.
2. The authors provide theoretical complexity analysis and convergence proof of the algorithms.
3. Experimental results indicate that CRONOS provides better or comparable performance compared to optimizers such as SGD, Adam, and AdamW, but hardly any tuning of hyperparameters.
4. This is the first time that convex networks have been applied to large datasets with competitive performance, which has great significance.

**Weaknesses:**

1.The symbols W1j, w2 are not consistent with their definition in line 77.
2. Figure 2 is not referenced in the paper.

**Questions:**

How is unsupervised training implemented on IMDC-DA?

**Limitations:**

The authors should conduct experiments on more datasets to test the effectiveness of the proposed algorithms.

---

> ### Author Rebuttal · Authors · 2024-08-06
>
> ** Weaknesses **
> 1. Thank you for pointing this out! We have fixed this in the revision.
>
> **Questions**
>
> We apologize for the typo and have corrected this for IMDB-DA. Thank you for raising the question! The Domain Adaptation experiments were aimed at minimizing the distribution shift between source and target domains. Our goal is to leverage the pre-trained high-level semantic knowledge in GPT2 and use the unlabeled IMDB data to help align this into our sentiment classification setting.
>
> We run one epoch of unlabeled IMDB data through a pre-trained GPT2 to predict the next token in the language modeling setting. This new checkpoint is then used for feature extraction and classification with our convex network and AdamW.
>
> This experiment was motivated by the idea of subspace alignment. Since pre-trained GPT2 has vast semantic knowledge, even minor alignment to an unlabeled dataset should be beneficial to the eventual downstream task. We are also practically motivated to explore challenging settings with unlabeled data, since this is readily prevalent in real world scenarios. We have added more numerical and experimental details for the IMDB-DA setting in the revision, and acknowledge this is an exciting area with much more work to be done!

---

### Author Rebuttal · Authors · 2024-08-06

We would like to sincerely thank all the reviewers for their thoughtful feedback and suggestions, which will greatly improve the submission.

In addition to our point-by-point response to each reviewer, we wish to underscore our contributions, describe the content of the rebuttal pdf, and address two key concerns shared by several reviewers.

**Contributions**

1. CRONOS is the first algorithm where convex neural networks have been successfully applied to large datasets with competitive performance.

2. CRONOS has strong theoretical convergence guarantees; we provide robust theoretical complexity analysis and convergence proof for the algorithm. This aims to bridge the gap between theory and practice in deep learning.

3. CRONOS-AM is the first algorithm to apply the convex reformulation to deeper networks and achieves excellent performance relative to standard optimizers. Our novel JAX implementation also provides the groundwork for efficient future experiments with fully leveraged GPU acceleration.

Ultimately our goal is to take a meaningful step towards an innovative and alternative paradigm in deep learning through convex networks. Although much further work remains to be done, we believe this paper opens up exciting new avenues in research with potential for real-world applications. We are highly enthusiastic about exploring the capability of convex networks in enhancing the efficiency and interpretability of deep learning in future work.

**Contents of Rebuttal pdf**
1. Figures (a) and (b) show the median performance of optimizers across a grid of learning rates with 95\% quantiles for CIFAR-10 and ImageNet. Please note that these plots now contain DAdapted-Adam (DAdam), and competitors no longer do poorly
on ImageNet due to the finer learning rate grid search. Tables 1 and 2 present the range in peak validation across the grid (except for CRONOS-AM and DAdam, which do not require a learning rate parameter).
CRONOS-AM outperforms DAdam on both tasks and performs comparably to the best-tuned first-order optimizer.
On ImageNet, Shampoo does best by a decent margin. We attribute this to Shampoo being an approximate second-order optimizer. Properly tuned, Shampoo may yield better performance than purely first-order optimizers like CRONOS-AM and Adam for certain tasks. The plots and the tables show competing methods exhibit a high degree of variance in the learning rate—poor selection can yield non-convergent behavior.
In contrast, CRONOS-AM does not exhibit these weaknesses and performs comparably to the best-tuned competitor.

2. Figures (c)-(f) compare CRONOS to AdamW on two GPT2 configurations. The first is (GPT2-NFT) which extracts embeddings from pre-trained GPT2 with no fine-tuning. The second configuration GPT2-FT extracts the embeddings after 1 epoch of finetuning with AdamW. Figures (c) and (e) compare CRONOS to tuned AdamW on time. In both cases, CRONOS reaches the best validation faster than AdamW.
Tuned AdamW fails to catch up for both configurations, with a particularly large gap ($\approx 4\%$) on GPT2-NFT.
Figures (d) and (f) plot several AdamW trajectories along with CRONOS against a number of epochs for AdamW and ADMM iterations for CRONOS. The more translucent the curve for AdamW indicates larger deviation from median trajectory. Both plots show AdamW is extremely sensitive to the learning rate selection. Tables 3 and 4 give the same information as Tables 1 and 2 for the GPT2 models.
Both support CRONOS's superior performance in terms of robustness to hyperparameter tuning, run time, and peak accuracy.

**Response to General Concerns**

1.  Several reviewers were concerned that in the original submission, most of the competing baselines performed poorly in the ImageNet experiment. This was due to the grid we searched over for the learning rate being inadequate. The original grid searched over learning rates ranging from $10^{-3.5}$ to $10^{-1.5}$. Notably, this grid includes the widely used default of $10^{-3}$ for methods like Adam, AdamW, and SGD employed in frameworks like PyTorch. Unfortunately, learning rates near the default perform poorly in this example, so a deeper search was required. Ultimately, we feel this underscores how difficult it can be to get the right learning rate.
    Our finer search ultimately resolves this issue - the competing methods no longer do poorly.
    CRONOS-AM still performs favorably relative to the competition.
    Please see the rebuttal pdf for more details and the new resulting plots, thank you.

2. There was a lack of a clear baseline for the original GPT2 experiments. We conducted 14 NLP experiments grouped into 3 categories (no train GPT2-NFT, 1 epoch fine tuning GPT2-FT, and unsupervised GPT2-DA). In each experiment, we benchmark against varying seed and learning rate sweeps of AdamW as a baseline. Please see the rebuttal pdf Figures (c) - (f) for a sample of these performance plots.
We select AdamW as our baseline since it is unanimously utilized in most NLP tasks.
Our results show that CRONOS outperforms tuned AdamW in both settings in terms of speed and accuracy. We will include all such results for the GPT2 experiments in the revision, along with a table of run-time comparisons.

Thank you once again to the reviewers for your insightful questions and valuable feedback. We look forward to engaging with you during the discussion phase!

---

### Decision · Program_Chairs · 2024-09-25

**Decision:**

Accept (poster)

**Comment:**

This paper introduces a novel way to train two-layer neural networks by a reformulation to a convex optimization problem. The CRONOS and CRONOS-AM algorithms are developped accordingly to solve the reformulated problem. The authors establish convergence guarantees and complexity analyses for the proposed algorithms. Experiment results on large datasets of image and language tasks to verify the effectiveness of the algorithm. Although the reviewers point out weaknesses of the paper such as insufficient explanations on the implementation in the experiments, all reviewers support accepting the paper after the rebuttal. The authors should add clarifications and discussions in the revised version of the paper according to the reviews and author-reviewer discussions.